# Antibody fine specificity correlates with protection from malaria for the RTS,S vaccine in young African children: A post hoc analysis of a phase IIb randomised controlled trial

Alessia Hysa[1,2], D. Herbert Opi[1,2,3], Joshua Waterhouse[1,3], Sandra Chishimba[1,2], Jessica L. Horton[1,4], Natalie Kingston[5], Hans J. Netter[6], David Wetzel[7], Michael Piontek[7,8], Gaoqian Feng[1,9], Jahit Sacarlal[10,11], Carlota Dobaño[12,13], Liriye Kurtovic[1,2,3�*], James G. Beeson[1,2,3�*]

1 Burnet Institute, Melbourne, Australia, 2 Department of Infectious Diseases and Department of Medicine, The University of Melbourne, Melbourne, Australia, 3 School of Translational Medicine and Department of Microbiology, Monash University, Melbourne, Australia, 4 Oxford Vaccine Group, Department of Paediatrics, University of Oxford, Oxford, England, United Kingdom, 5 Astbury Centre for Structural Molecular Biology, School of Molecular & Cellular Biology, Faculty of Biological Sciences, University of Leeds, Leeds, United Kingdom, 6 School of Science, RMIT University, Melbourne, Australia, 7 ARTES Biotechnology GmbH, Langenfeld, Germany, 8 Laboratory of Plant and Process Design, Technical University of Dortmund, Dortmund, Germany, 9 National Vaccine Innovation Platform, NHC Key Laboratory of Antibody Technique, Jiangsu Key Laboratory of Pathogen Biology, Department of Pathogen Biology, Nanjing Medical University, Nanjing, Jiangsu, China, 10 Centro de Investigação em Saúde de Manhiça, Maputo, Mozambique, 11 Faculdade de Medicina, Universidade Eduardo Mondlane (UEM), Maputo, Mozambique, 12 ISGlobal, Hospital Clínic Universitat de Barcelona, Barcelona, Catalonia, Spain, 13 SGlobal, Hospital Clínic Universitat de Barcelona, and Centro de Investigación Biomédica en Red de Enfermedades Infecciosas (CIBERINFEC), Barcelona, Spain

উ These authors contributed equally to this work and are co-senior authors on this work.
* liriye.kurtovic@burnet.edu.au (LK); james.beeson@burnet.edu.au (JGB)

## Abstract

### Background

The RTS,S/AS01 malaria vaccine was recently approved for implementation in children, but only provides modest and short-lived efficacy against malaria. RTS,S targets a portion of the *Plasmodium falciparum* (*Pf*) circumsporozoite protein (CSP), comprising the central NANP-repeat region and C-terminal domain. Mechanisms of immunity and correlates of protection for the RTS,S vaccine are not well defined, hindering progress towards generating highly effective CSP-based vaccines.

### Methods and findings

We investigated epitope specificity and cross-reactivity of vaccine-induced antibodies to six peptides representing CSP epitopes in the N-terminal and central NANP-repeat region. We evaluated antibody reactivity in preclinical mouse vaccine studies, among CSP-specific monoclonal antibodies (mAbs), and in a large RTS,S phase IIb clinical trial in young children 1–4 years old (*n* = 735).

the Creative Commons Attribution License, which permits unrestricted use, distribution, and reproduction in any medium, provided the original author and source are credited.

**Data availability statement:** A de-identified database is provided as a supplementary S1 Data file. Databases that were analysed in the preparation of this paper and presented in figures and tables are available (de-identified format) by contacting Nadine Barnes (nadine. barnes@burnet.edu.au; Research Integrity Office of Burnet Institute, AU) from the date of publication of this paper, for non-commercial purposes and pending the completion of a written agreement.

**Funding:** This work was supported by the National Health and Medical Research Council (NHMRC) of Australia (Investigator Grant 2033320 and Synergy Grant 2018654 to JGB; https://www.nhmrc.gov.au/), the Thrasher Research Fund (Early Career Award to LK; https://www.thrasherresearch.org/), the CASS Foundation (Medicine/Science Grant to LK; https://www.cassfoundation.org/), and Research Training Program Scholarships, Australian Government (AH, JW, SC, JH; https://www.education.gov.au/research-block-grants/research-training-program). AH, DHO, JW, SC, GF, LK, and JGB are members of the Australian Centre for Research Excellence in Malaria Elimination, funded by the NHMRC (Synergy Grant 2018654 to JGB). AH, DHO, JW, SC, LK, and JGB study and work at the Burnet Institute which is supported by the NHMRC for Independent Research Institutes Infrastructure Support Scheme and the Victorian State Government Operational Infrastructure Support. CD works at ISGlobal which is a member of the CERCA Program, Generalitat de Catalunya. CD and JS work at the Manhiça Health Research Centre which receives core funding from the Spanish Agency for International Cooperation and Development (AECID). The following salaries were supported by grant funding: JGB, LK, DHO, GF, (NHMRC Investigator Grant 2033320 to JGB). The funders had no role in study design, data collection and analysis, decision to publish, or preparation of the manuscript.

**Competing interests:** I have read the journal's policy and the authors of this manuscript have the following competing interests: The RTS,S phase IIb clinical trial [NCT00197041] was previously funded by GlaxoSmithKline S.A. who was provided the opportunity to review a

The preclinical mouse vaccine studies and CSP-specific mAbs were used to initially evaluate IgG responses to the six peptides. Mice immunised with the central NANP-repeat region had IgG with cross-reactivity to an epitope in the N-terminal region. Additionally, we demonstrated that a single CSP-specific mAb could display cross-reactivity to several CSP epitopes. Through post hoc quantification and analysis of antibody responses in the RTS,S phase IIb clinical trial, we found that a subset of children generated IgG with specificity for a short NANP-repeat epitope (NANP$_2$; amino acid sequence: NANPNANP) and cross-reactivity to an N-terminal epitope (J1; amino acid sequence: KQPADGNPDP-NANPN). Notably, children with high IgG responses to NANP$_2$ and J1 had a significantly reduced risk of clinical malaria, compared to children with low responses (IgG to NANP$_2$ (aHR: 0.838 (95% CI [0.716, 0.981]; $p = 0.028$)) and J1 (aHR: 0.718 (95% CI [0.611, 0.844]; $p < 0.001$)), and these responses were also associated with higher antibody Fc-mediated functional activities. We have evaluated NANP$_2$ and J1 as immunological correlates of protection in one phase IIb cohort, additional studies in other RTS,S and R21 cohorts will be important to further confirm our findings.

## Conclusions

These findings reveal promising new correlates of protection for RTS,S and new insights to inform the development of next-generation malaria vaccines.

## Author summary
### Why was this study done?

- Annually, over 240 million malaria cases occur, resulting in over 600,000 deaths, predominantly in young children under the age of 5.

- The RTS,S malaria vaccine is being implemented among young children in some regions of Africa. However, it provides only moderate and short-lived protection. To substantially reduce the global malaria burden, vaccines with higher efficacy and longevity are needed.

- To generate highly protective vaccines, knowledge of how malaria vaccines mediate protection is required. Furthermore, established correlates of protection for RTS,S are lacking, restricting progress and assessment of vaccine effectiveness.

### What did the researchers do and find?

- This study aimed to identify the specific parasite targets of protective immune responses in children vaccinated with the RTS,S malaria vaccine.

- Initially, mouse vaccine studies were used to establish that the study protocol could detect specific immune responses of interest.

draft of this manuscript for factual accuracy, but the authors are solely responsible for the final content and interpretation. MP is currently the managing director and DW is a former employee of ARTES Biotechnology GmbH and JGB is an Academic Editor on PLOS Medicine Editorial Board. The other authors declare that no competing interests exist.

**Abbreviations:** ARRIVE, animal research: reporting of in vivo experiments. AUC, area under the curve; CIconfidence intervalCSP, circumsporozoite protein; ELISA, enzyme-linked immunosorbent assay; HRhazard ratiomAbs, monoclonal antibodies; ODoptical densityPCA, principal component analysis; PFplasmodium falciparumSD, standard deviation; STROBE, Strengthening the Reporting of Observational Studies in Epidemiology; VLP, virus-like particles.

- Subsequently, immune responses were analysed in blood samples of vaccinated children to identify specific targets of antibodies that were linked with protection from malaria.

- The study revealed that vaccinated children who were protected against malaria had strong immune responses to two specific malaria parasite targets, but these responses were only present in a minority of children.

## What do these findings mean?

- The study identified specific parasite targets of RTS,S-induced antibodies that are associated with protection from malaria and should be considered in the design of next-generation malaria vaccines.

- The specific responses identified provide correlates of protection that can be used in future clinical trials to evaluate protective immune responses. Further studies are required to independently confirm the protective associations identified in this study.

## Introduction

Malaria continues to pose an immense global health burden with over 200 million cases and 600,000 deaths reported in 2023 [1]. Progress in reducing the global malaria burden has stalled in the past decade, highlighting the need for highly efficacious and long-lasting vaccines. Two malaria vaccines, RTS,S/AS01 and R21/Matrix-M, have been recommended for young children (first dose around 6 months of age) in areas with moderate to high malaria transmission, alongside other established antimalarial control measures [1]. These vaccines are based on the same circumsporozoite protein (CSP) antigenic construct and although they differ in their formulations, they are currently deemed to be comparable in efficacy by the World Health Organisation [2]. RTS,S and R21 efficacy varies among populations, malaria transmission settings and across age groups, being highest in young children. In a phase III trial, RTS,S provided 55% efficacy against malaria in children for a period of 12 months, which varied by geographic location, and waned within 18 months of vaccination [3]. Consequently, a booster dose is recommended after 12–18 months. In a phase III trial of R21, efficacy was 68%–75% for a period of 12 months which varied by malaria transmission epidemiology and was lower in older compared to younger children (18–36 months versus 6–18 months) [4]. To enable the development of more efficacious vaccines, new insights into RTS,S and R21-induced immune responses linked with protective efficacy and the establishment of robust correlates of protection are needed.

RTS,S and R21 target the major surface antigen expressed on *Plasmodium falciparum* sporozoites known as the CSP and aim to prevent hepatocyte infection and subsequent progression into blood stage parasitemia that results in clinical illness. CSP comprises the (i) N-terminal region with "Region I" thought to be the site for proteolytic cleavage of CSP required for hepatocyte attachment, (ii) central repeat region composed of 37–49 major NANP (Asparagine-Alanine-Asparagine-Proline) and ~4 minor NVDP (Asparagine-Valine-Aspartic Acid-Proline) repeat motifs, and (iii) C-terminal region, which includes a thrombospondin-like type I repeat domain and T-cell epitopes (Fig 1A). CSP also includes a highly conserved junction epitope that links

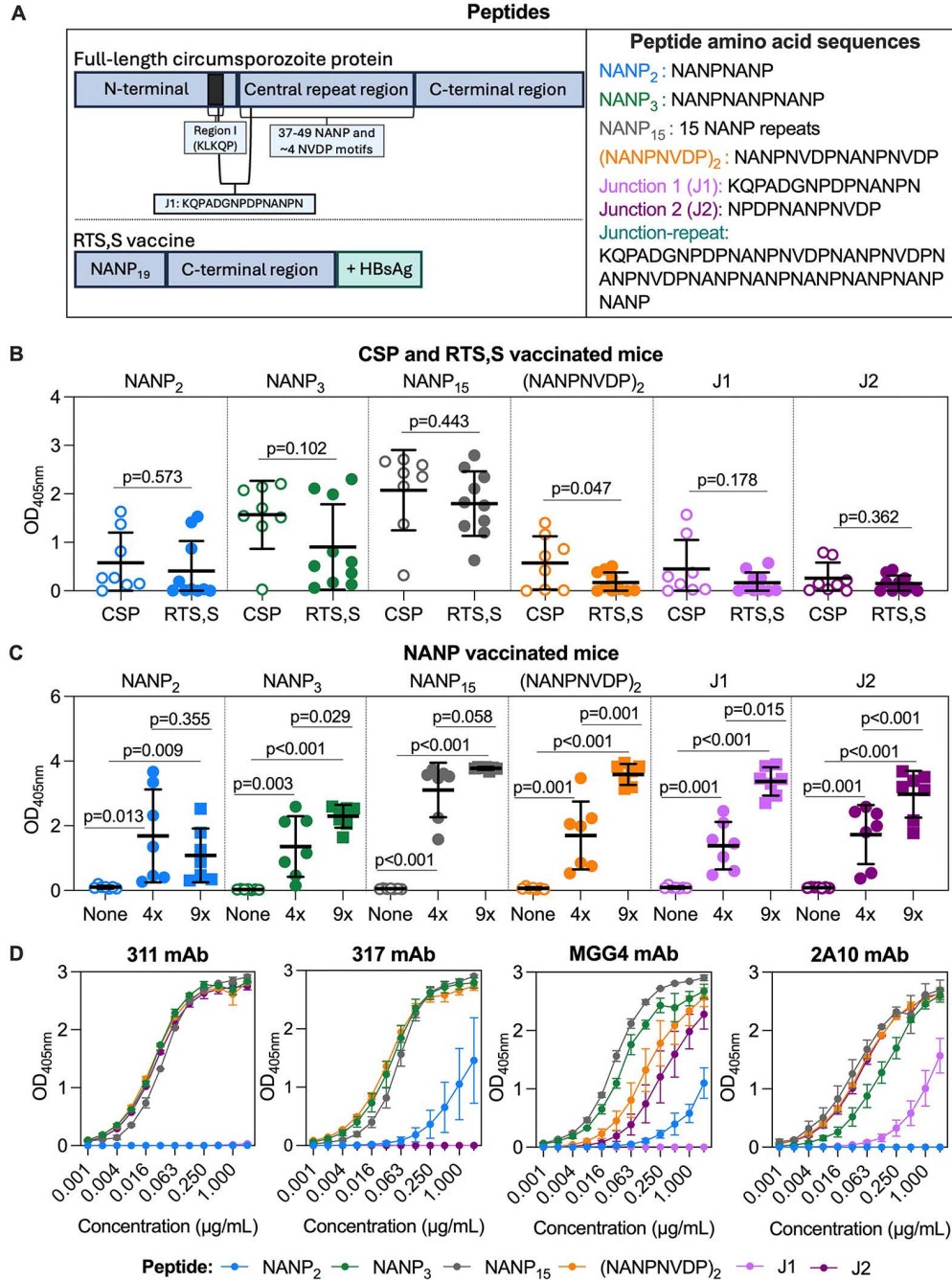

**Fig 1. IgG specificity and cross-reactivity to CSP major repeat, minor repeat, and junction epitopes in mouse vaccine studies and human monoclonal antibodies. A)** Full-length *Plasmodium falciparum* circumsporozoite protein (CSP) comprises the N-terminal (with "Region I"), central repeat region that includes multiple NANP and NVDP-repeat motifs, C-terminal region, and a junction epitope (KQPADGNPDPNANPN) that overlaps the N-terminal and central repeat regions. The RTS,S vaccine construct includes 19 NANP-repeats and the C-terminal region fused with human hepatitis B surface antigen (HBsAg) and co-expressed with excess HBsAg to form virus-like particles (VLPs). **B)** Mice were vaccinated with full-length CSP (open circle, $n = 8$) or our RTS,S-like construct (closed circle, $n = 10$) and evaluated for IgG to NANP$_2$, NANP$_3$, NANP$_{15}$, (NANPNVDP)$_2$, J1, and J2 peptides representing epitopes of the CSP antigen by ELISA. Sera collected after vaccination were tested at a 1/4000 dilution and the mean of duplicates is shown along with the group mean (centre line) and standard deviation (whiskers). An unpaired $t$ test was performed, and corresponding p-values are shown. **C)** Mice were vaccinated with a VLP expressing no antigen (open circle, $n = 7$), 4× NANP-repeats (closed circle, $n = 7$), or 9× NANP -repeats (closed square, $n = 7$) and evaluated for IgG to peptides representing CSP epitopes by ELISA. Sera collected after vaccination were tested at a 1/100 dilution and the

mean of duplicates is shown along with the group mean (centre line) and standard deviation (whiskers). An unpaired *t* test was performed, and corresponding *p*-values are shown. **D)** Human CSP-specific mAbs 311, 317, MGG4, and 2A10 were tested between 0.001 and 2 µg/mL for IgG to peptides representing CSP epitopes by ELISA. mAbs 311 and 317 were isolated from an individual vaccinated with RTS,S, MGG4 was isolated from an individual vaccinated with whole irradiated sporozoites (*Pf*SPZ), and 2A10 is a humanised mAb isolated from mice vaccinated with *Pf*SPZ. The mean and range of two independent experiments are shown.

the N-terminal and central repeat regions and is characterised by the presence of NPDP (Asparagine-Proline-Aspartic Acid-Proline)/NVDP amino acid motifs [5–13]. Notably, RTS,S and R21 include a truncated form of CSP comprising the entire C-terminal region and 19 NANP-repeat motifs, fused with the hepatitis B virus-derived surface antigen (HBsAg) to self-assemble as virus-like particles (VLP) [14–18]. The vaccine construct lacks the N-terminal region, junction epitope, and minor NVDP-repeat motifs of CSP, which may be significant targets of immunity.

RTS,S vaccine immunity is primarily mediated by antibodies and CD4+ helper T-cells to support antibody generation and memory B cell responses [19–24]. The functional activities of vaccine-induced antibodies include complement fixation against sporozoites and interacting with Fc-receptors to promote phagocytosis and antibody-dependent killing of sporozoites [19,25–28]. Vaccine immune responses are currently primarily measured as IgG magnitude to an antigen comprised of 30 NANP-repeats. However, this response only moderately or inconsistently correlates with protection against malaria in RTS,S clinical trials in children, demonstrating variable protective associations in some studies but not all [3,29–38] (summarised in S1 Appendix, Table A). Quantifying antibodies to a long NANP-repeat sequence may not capture the fine specificity of responses that may be important for, or the best correlate of, protective efficacy [39]. The intrinsically disordered structure of the central repeat region is such that it is flexible and dynamic, forming various epitopes composed of repeated NANP and/or NVDP motifs [40]. Although the minor central repeat NVDP motif and NPDP junction epitope motif are excluded from the vaccine construct, they share amino acid similarities to the major NANP-repeat motif included in RTS,S. Human monoclonal antibodies (mAbs) that bind NANP-repeats have been shown to cross-react with NPDP and NVDP minor repeats [41–44]. However, there are no published data on polyclonal antibody responses to short NANP-repeats, minor NVDP motif, or junctional regions of CSP in children vaccinated with RTS,S who reside in malaria endemic regions, and whether these correlate with vaccine efficacy. This is a critical knowledge gap as at-risk children are the primary target group for vaccination.

Given the potential importance of epitope specificity and the lack of robust immunological correlates of protection, evaluating whether RTS,S vaccine-induced antibodies can recognise epitopes including NVDP- and NPDP-repeats and epitopes consisting of different NANP-repeat lengths may reveal insights into antibody specificity that correlate with vaccine efficacy against natural exposure to malaria. Here, we evaluated the fine specificity of vaccine-induced IgG response in established mouse models using different CSP-based antigens including those contained in the RTS,S and R21 vaccines. We investigated IgG specificities in young African children vaccinated with RTS,S as part of a post hoc analysis of a phase IIb clinical trial. We evaluated whether the fine specificity of IgG correlated with protection against malaria and analysed the acquisition of antibodies with different specificities among young children.

## Methods

### Ethics statement

Written informed consent was provided by the parents or guardians of all participants. Ethics approvals were obtained from the Mozambican National Health and Bioethics Committee, Hospital Clinic of Barcelona Ethics Committee, PATH Research Ethics Committee and Human Research and Ethics Committee at the Alfred Hospital (protocol 174/18). The use of samples in this study is covered by the original consent agreement. Additional information regarding the ethical, cultural, and scientific considerations specific to inclusivity in global research is included in the Supporting Information (S1 Checklist). Animal immunisations were performed at Walter and Eliza Hall Institute (WEHI, AU) animal facility (ethics approval number: 2020.019).

## Animal immunisations

We evaluated the epitope specificities of vaccine-induced antibodies in mice immunised with full-length CSP, an RTS,S-like vaccine, and a vaccine that includes only NANP-repeat motifs. Firstly, we leveraged existing samples from a study we previously conducted where mice were vaccinated with a truncated form of CSP (same as the RTS,S vaccine) expressed as VLPs using the duck HBsAg ($n=10$) presentation platform METAVAX [45], which is similar to HBsAg, and was produced using *Hansenula polymorpha* at ARTES Biotechnology (Germany). Another group of mice were vaccinated with recombinant full-length CSP ($n=8$) [46]. Secondly, to assess antibodies induced by the NANP-repeat motif alone, we leveraged existing samples from a study we previously conducted where mice were vaccinated with VLPs (using human HBsAg) expressing $NANP_4$ ($n=7$), $NANP_9$ ($n=7$), or as a negative control no malaria antigen (HBsAg only, $n=7$), as previously described [47]. In these VLPs, the NANP-repeat sequences were inserted into the surface-exposed loop 2 of HBsAg. For a detailed description of the mouse vaccine studies, please refer to S1 Appendix. The studies are reported as per the Animal Research: Reporting of In Vivo Experiments (ARRIVE) Guidelines (S2 Checklist).

## Monoclonal antibodies

CSP-specific mAbs were expressed as recombinant human IgG1 in mammalian HEK293 cells using established approaches and tested in a 2-fold dilution series between 0.001 and 2 µg/mL [26,36]. These included mAbs 311 and 317, which were previously isolated from a malaria-naïve participant immunised with RTS,S in a phase IIa controlled human malaria infection study [40]. We also evaluated mAb MGG4 isolated from a malaria-exposed participant vaccinated with live-attenuated sporozoites [43], and mAb 2A10 isolated from a mouse vaccinated with live-attenuated sporozoites [48]. All mAbs have been previously characterised and are known to recognise the major NANP-repeat motif of CSP. The sequences of these mAbs were previously published, enabling us to produce them for use in our study and the mAbs provided a representative selection relevant to our questions and to confirm the performance of our assays.

## Study participants

We evaluated responses in children aged 1–4 years recruited within a 10 km radius of the Manhiça district (Maputo Province, Mozambique; $n=1,605$) for the RTS,S/AS02 phase IIb, double-blind, randomised controlled clinical trial in 2004 (ClinicalTrials.gov registry number NCT00197041). Within the study period (April 2003 and May 2004) the area experienced perennial malarial transmission with pronounced seasonality. Further details of the trial protocol can be found in Alonso and colleagues [30]. The primary outcome was vaccine efficacy against clinical *P. falciparum* malaria (axillary temperature ≥37.5 °C and asexual parasitaemia ≥2,500 parasites per µL), which was 29.9% (95% CI [11.0,44.8%]) in the first 6 months; in the control group, there were 159 recorded events per 302.9 person-years at risk within the 6 month follow-up period for the primary case definition. We evaluated serum samples collected in all available children who completed the primary vaccination series and received three doses of RTS,S/AS02 ($n=735$) at months 0, 1, and 2 of the study. As a control, we also evaluated serum samples collected in available children ($n=99$) from the trial immunised with three doses of non-malaria vaccines (pneumococcal conjugate, *Haemophilus influenzae* type B or paediatric hepatitis B vaccines, depending on their age) at months 0, 1, and 2 of the study. Serum samples were collected 30 days post-primary vaccination series at month 3 of the study period and tested in immunoassays. Due to limited sera availability, we initially generated pooled samples to screen antibody reactivity to each peptide prior to testing all available samples from individual children in the RTS,S ($n=735$) and control ($n=99$) vaccine groups (S1 Appendix, Fig A). We generated the pooled samples by randomly selecting children identified as malaria-free ($n=50$) or malaria-positive ($n=50$) over 18 months post-vaccination. As a negative control, we also pooled samples from a random selection of children in the control non-malaria vaccine group ($n=50$). To evaluate correlates of protection, we included samples for the subset of children with clinical data available ($n=646$ for RTS,S and $n=84$ for control). For functional immunoassays, we only had sera from

a sub-group of children immunised with RTS,S ($n = 461$) and the control vaccine ($n = 23$) available to test with the Junction-repeat peptide (S1 Appendix, Fig E). A summary of our study design is outlined in Fig A of S1 Appendix. This study is reported as per the Strengthening the Reporting of Observational Studies in Epidemiology (STROBE) guidelines, and a checklist is included in the Supporting Information (S3 Checklist).

## Antigens

The following peptides were synthesised to represent epitopes within the *P. falciparum* CSP for use in immunoassays: various lengths of the major NANP-repeat ($NANP_2$ and $NANP_3$) based on prior findings that the minimal antibody epitope is approximately 1.5–2.5 repeats in length [40]; the minor repeat $(NANPNVDP)_2$ and junction epitopes (Junction 1 (J1), and Junction 2 (J2)) that are not included in RTS,S (Mimotopes, AU, Table 1 and Fig 1A). Note that shorter peptides (<20 amino acids in length) were biotinylated to enhance coating for IgG immunoassays. It was technically challenging to perform functional immunoassays with the short, biotinylated peptides; therefore, we synthesised a longer peptide comprising NANP, NVDP and NPDP motifs (Junction-repeat peptide) to measure antibody functional activity to these motifs (sequences shown in Table 1 and Fig 1A). We previously determined that RTS,S-induced IgG were highly reactive to 15 NANP-repeats, therefore, the long $NANP_{15}$ peptide was also included to represent the central repeat region of CSP (LifeTein, USA) [19].

## Immunoassays to detect antibody binding and functional responses

We evaluated IgG binding to each peptide by enzyme-linked immunosorbent assay (ELISA) as previously described [36]. We also evaluated the ability of antigen-specific antibodies to mediate functional activities, including C1q complement fixation and FcγRIII binding, using established immunoassays [36]. Further details can be found in S1 Appendix.

## Statistical analysis

Descriptive analyses were performed using GraphPad Prism 10.4.0 (GraphPad Software). Antibody responses among independent vaccine groups were compared using an unpaired *t* test in preclinical studies and the Mann–Whitney *U* test in clinical studies. Antibody responses for matched participants in the clinical study were compared using a Wilcoxon signed-rank test. We performed parametric statistical testing for the preclinical studies as the mice were genetically identical and raised in the same pathogen-free conditions and therefore assumed to have a normal data distribution. We used non-parametric statistical tests for the clinical study as vaccine responses in African children have been previously shown to be heterogenous and not normally distributed [36]. The relationship between IgG specificity to different peptides

**Table 1. Synthetic peptides representing the central repeat and junction epitopes of CSP.**

| Peptide Name | Peptide Amino Acid Sequence |
|---|---|
| $NANP_2$* | NANPNANP |
| $NANP_3$* | NANPNANPNANP |
| $NANP_{15}$ | NANPNANPNANPNANPNANPNANPNANPNANP-NANPNANP NANPNANPNANPNANPNANP |
| $(NANPNVDP)_2$* | NANPNVDPNANPNVDP |
| Junction 1 (J1)* | KQPADGNPDPNANPN |
| Junction 2 (J2)* | NPDPNANPNVDP |
| Junction-repeat | KQPADGNPDPNANPNVDPNANPNVDPNAN-PNVDPNANPNANP NANPNANPNANPNANP |

*Peptide conjugated to biotin.

within clinical studies was assessed using Spearman's rank correlation coefficient (r or Rho) which assumes a monotonic relationship. The area under the curve (AUC) was calculated for RTS,S pools that were tested at a 2-fold dilution series between 1/250 and 1/64,000 dilution. We evaluated the association between high and low IgG response to $NANP_{15}$, $NANP_2$ and J1 antigens (based on the median optical density (OD)) and time to first malaria episode recorded until ~1.5 years (18 months) post vaccination using Kaplan–Meier curves and the Log rank test (Mantel-Cox). A Cox proportional-hazards model was performed using Stata (version 15) to determine the adjusted hazard ratio (aHR; adjusted for age) of first malaria episode for an increase in antibody response normalised by the standard deviation (SD), as previously described [36]; we evaluated the proportional hazards assumption test ($p > 0.05$) and confirmed the assumptions were met. Age and sex were considered as potential confounders, *a priori*, but only age was associated with vaccine responses and was included in the final adjusted model (S1 Appendix, Fig A). In prior published analyses of antibody associations with protection, only age was identified as a potential confounder of the association between vaccine-induced antibodies and subsequent risk of malaria [36]. The statistical analysis plan for the present study followed the time-to-event analysis approach reported for the original clinical trial and a previously published analysis of the associations between antibodies and malaria events [30,36]. The analysis was designed to evaluate the hypothesis that specific IgG responses induced by the vaccine, measured at month 3, would be associated with risk of malaria during the follow-up period. To evaluate the relationship between IgG responses to $NANP_{15}$, $NANP_2$ and J1 in RTS,S vaccinated children we performed principal component analysis (PCA) and cluster analysis in R (v4.3.1, 'Beagle Scouts') using RStudio (v2023.6.2.561). We specifically used (*'fviz_cluster()'*) function (*factoextra* v1.0.7 R package, repositories: CRAN & GitHub). For this analysis, antibody data were log-transformed and normalised, participants were clustered using a hierarchical k-means approach (*hkmeans*) and a hybrid clustering approach was selected to produce more robust clusters relative to traditional k-means clustering. Hierarchical clustering was performed using Euclidean distance and Ward algorithm. K-means clustering was performed using Hartigan–Wong algorithm after 10 iterations. *P* values are reported for all statistical tests; no adjustments for multiple comparisons were performed.

## Results

### RTS,S and CSP-based vaccines induce IgG to minor repeat and junction epitopes in mouse models

We evaluated the extent to which vaccination with the RTS,S antigen induced antibodies that recognised NANP-repeat sequences of different lengths, as well as the minor NVDP-repeat and junction epitopes that are excluded from the vaccine construct. We performed initial studies in mice using an ELISA to capture fine specificity and cross-reactivity of vaccine-induced antibodies to short, biotinylated peptides. The mouse samples also avoided the potential confounding effect of prior exposure to malaria which is present in vaccinated children from endemic regions. Mice were vaccinated with a VLP expressing the RTS,S vaccine construct ($n = 10$) or full-length recombinant CSP for comparison ($n = 8$), and sera collected after the third dose were used to quantify IgG to peptides representing different CSP epitopes (Fig 1A). These included various lengths of the major NANP-repeat ($NANP_2$, $NANP_3$, and $NANP_{15}$) and the minor NVDP-repeat and junction epitopes that are excluded from RTS,S (($NANPNVDP)_2$, J1, and J2). All RTS,S-vaccinated mice were seropositive for IgG to the major NANP-repeat epitope (presented as $NANP_{15}$), and 90% were also positive for IgG to a shorter epitope comprised of three major repeats ($NANP_3$) (Fig 1B). However, only a subset of mice had antibodies that could bind the short $NANP_2$ epitope (40%). A moderate proportion of RTS,S-vaccinated mice were seropositive to the minor repeat $(NANPNVDP)_2$ (40%), and junction peptides, J1 (40%) and J2 (50%), despite these epitopes being excluded from the RTS,S vaccine construct. Seropositivity threshold was defined as $OD = 0.1$ based on the overall mean + 2SD response of pre-vaccine samples to each peptide. IgG responses to all peptides were comparable between the RTS,S and full-length CSP vaccine groups, except for $(NANPNVDP)_2$, which was higher with CSP vaccination ($p = 0.047$).

We performed a follow-up study of mice vaccinated with a VLP expressing only an NANP sequence of ×4 ($n = 7$) or ×9 ($n = 7$) repeats in length, or no malaria antigen as a negative control ($n = 7$; Fig 1C). Mice in the $NANP_4$ and $NANP_9$ groups

had significant IgG to all peptides, including those representing the minor NVDP-repeat and junction epitopes, compared to the no-antigen control group ($p < 0.05$ for all tests). IgG responses tended to be higher in the $NANP_9$ group than the $NANP_4$ group, and it was notable that vaccination with $NANP_9$ appeared to be significantly better at inducing antibodies to the junction epitopes (J1 and J2) and minor repeats (($NANPNVDP)_2$) ($p < 0.05$ for all tests). Collectively, these data support that vaccination with the RTS,S vaccine construct, and more specifically, NANP-repeat sequences, can induce IgG that binds to minor NVDP-repeat and junction epitopes of the CSP antigen.

To further assess antibody specificity, we evaluated the reactivity of CSP-specific mAbs known to bind NANP-repeats against our peptide panel (Fig 1D). Each peptide included in the study was recognised by at least one of the mAbs tested. mAbs 311 and 317, derived from an RTS,S vaccinated adult, had strong reactivity to $NANP_3$ and $NANP_{15}$ peptides but also demonstrated cross-reactivity to the ($NANPNVDP)_2$ minor repeat peptide. mAb 311 also recognised the NPDPNAN-PNVDP sequence of the J2 peptide, whereas mAb 317 recognised the $NANP_2$ peptide at higher concentrations (>0.25μg/mL). In comparison, mAb MGG4, derived from an individual immunised with live-attenuated sporozoites, demonstrated broad reactivity to all peptides including NANP, NPDP, and NVDP motifs except the J1 peptide (KQPADGNPDPNANPN). The humanised mAb 2A10 originally derived from a mouse vaccinated with live-attenuated sporozoites recognised all peptides, including J1, but not the short $NANP_2$ peptide. Results indicate that a single antibody can have varying degrees of cross-reactivity profiles, recognising the major and minor repeats, and junctional regions of CSP. These analyses of preclinical vaccination samples and mAbs established the ability of our assays to detect differences in antibody specificities to specific epitopes for use in analysis of paediatric RTS,S clinical trials.

## Specificity and cross-reactivity of IgG in young African (Mozambique) children vaccinated with RTS,S

We evaluated the fine specificity and cross-reactivity of antibodies induced by RTS,S vaccination in African children (aged 1−4 years) using our established assays. Due to limited sample volume, we first generated pooled samples to screen peptide reactivity, including a pool of vaccinated children who remained malaria free ($n = 50$) and those who had malaria events during follow-up ($n = 50$), and children who received the control non-malaria vaccine ($n = 50$; Fig 2). We observed IgG specificity and cross-reactivity to all peptides in pooled samples from RTS,S-vaccinated children (OD > 0.2 at a 1/250 dilution) compared to the pooled sample from children in the control vaccine group which displayed no reactivity (OD < 0.2 at a 1/250 dilution). We found higher reactivity to the $NANP_2$ and J1 peptides in the pool of children who remained malaria-free compared to those who developed malaria (3.5× and 2.1× higher AUC units, respectively), suggesting they may be clinically important antibody targets. To further explore the relationship between the number of NANP-repeats in a sequence and IgG binding, we tested a subset of samples from children vaccinated with RTS,S (month 3, $n = 40$) for IgG reactivity to $NANP_2$, $NANP_3$ and $NANP_{15}$ in parallel assays (S1 Appendix, Fig B). IgG reactivity to $NANP_3$ was greater than $NANP_2$ but was very similar to IgG to the longer $NANP_{15}$ antigen. Furthermore, $NANP_3$ and $NANP_{15}$ reactivity were highly correlated (rho = 0.951 (95% CI [0.907,0.975]); $p < 0.001$), suggesting that these antigens measure or represent the same antibody response type (S1 Appendix, Fig B). In contrast, only a subset of children had reactivity to $NANP_2$, which was only moderately correlated with $NANP_3$ (rho = 0.269 (95% CI [−0.055,0.543]); $p = 0.093$) and $NANP_{15}$ (rho = 0.375 (95% CI [0.063,0.621]); $p = 0.017$). These findings suggest that quantifying IgG to $NANP_2$ may be valuable in defining different response types among vaccinated children.

We further evaluated IgG to $NANP_2$ and J1 peptides in individual samples from children who received RTS,S ($n = 735$) or the control non-malaria vaccine ($n = 99$). For comparison, we also included data for IgG to the long $NANP_{15}$ peptide, which is known to be strongly recognised by RTS,S-induced IgG (Fig 3A) [36]. We found significantly higher magnitudes of IgG to $NANP_{15}$, $NANP_2$ and J1 peptides in children given RTS,S compared to control vaccine ($p < 0.001$ for all tests). Interestingly, IgG responses to J1 and $NANP_{15}$ peptides were consistently lower in older children (average, 41.6 months; range, 24.0–59.8 months) compared to younger children (average, 17.7 months; range, 12.3–23.9 months; $p < 0.001$ for both antigens; Fig 3B). On the other hand, there was only a weak age effect for IgG to $NANP_2$ ($p = 0.053$). There was no difference in IgG

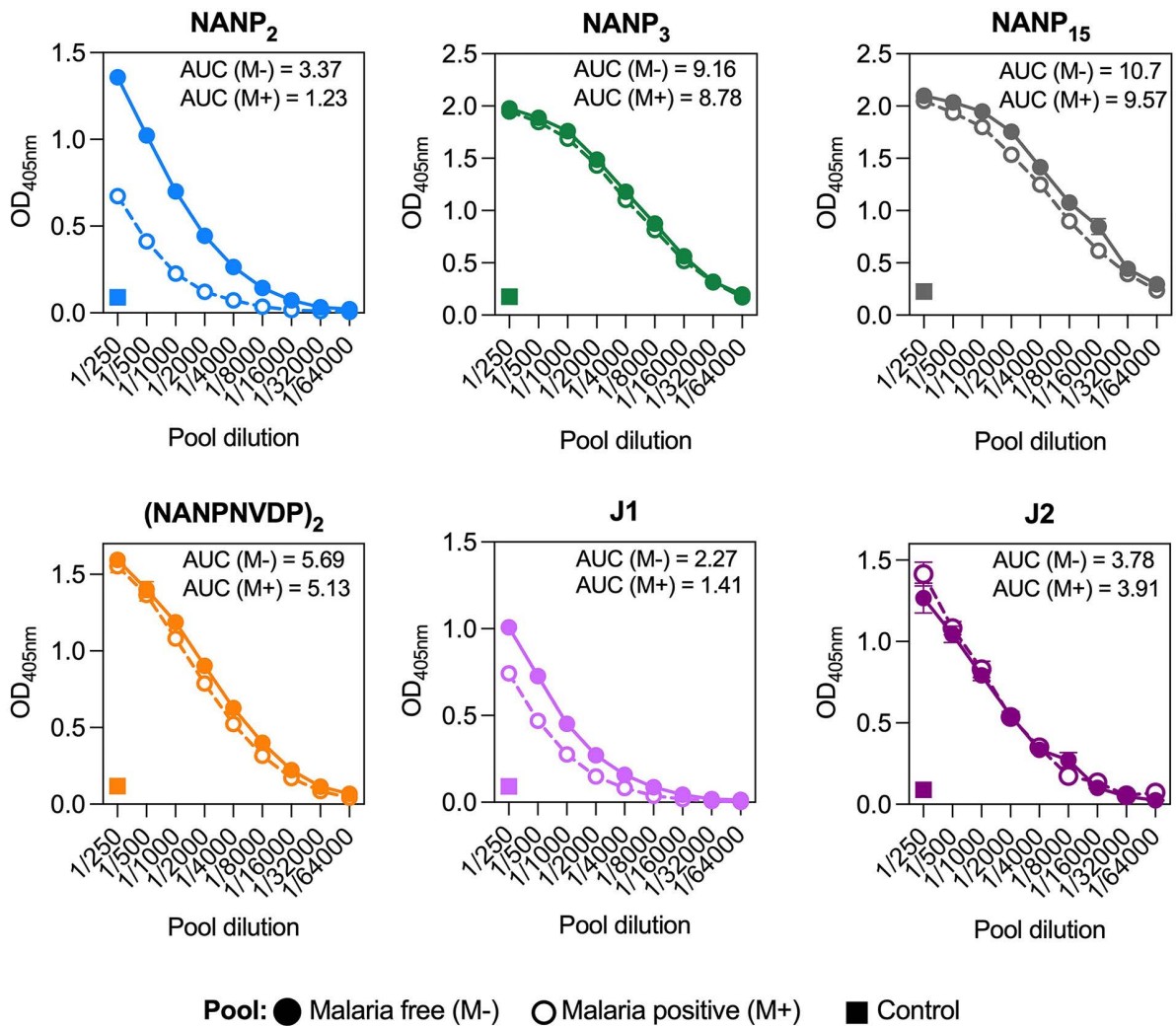

**Fig 2. IgG to CSP major repeat, minor repeat, and junction epitopes in pooled samples from RTS,S vaccinated children.** Pooled sera from RTS,S vaccinated children (study month 3 time point) who remained malaria free (M−, closed circle, $n = 50$) or were malaria positive (M+, open circle, $n = 50$) during follow-up were tested between a 1/250 and 1/64,000 dilution for IgG to $NANP_2$, $NANP_3$, $NANP_{15}$, $(NANPNVDP)_2$, J1, and J2 peptides by ELISA. Pooled sera from children administered a control non-malaria vaccine (closed square, $n = 50$) was also tested at a 1/250 dilution. The mean and range of duplicates are shown along with the area under the curve (AUC) for malaria free (M−) and malaria positive (M+) pools.

reactivity for any peptide by sex ($p > 0.05$ for all tests; Fig 3C) or malaria exposure based on IgG reactivity against *P. falciparum* merozoites ($p > 0.05$ for all tests; S1 Appendix, Fig C). There was a moderate correlation between the avidity of IgG to $NANP_{15}$ and IgG magnitude to J1 peptide, suggesting higher avidity NANP-repeat antibodies are somewhat more likely to have cross-reactive binding activity ($n = 36$, rho = 0.449 (95% CI [0.131,0.683]); $p = 0.006$; S1 Appendix, Fig D).

**IgG fine specificity and cross-reactivity are associated with protection against clinical malaria in young children vaccinated with RTS,S**

We evaluated the association between IgG reactivity to the different peptides and time to first clinical malaria episode in children who received all three doses of RTS,S and had complete follow-up data over approximately 1.5 years after

PLOS Medicine

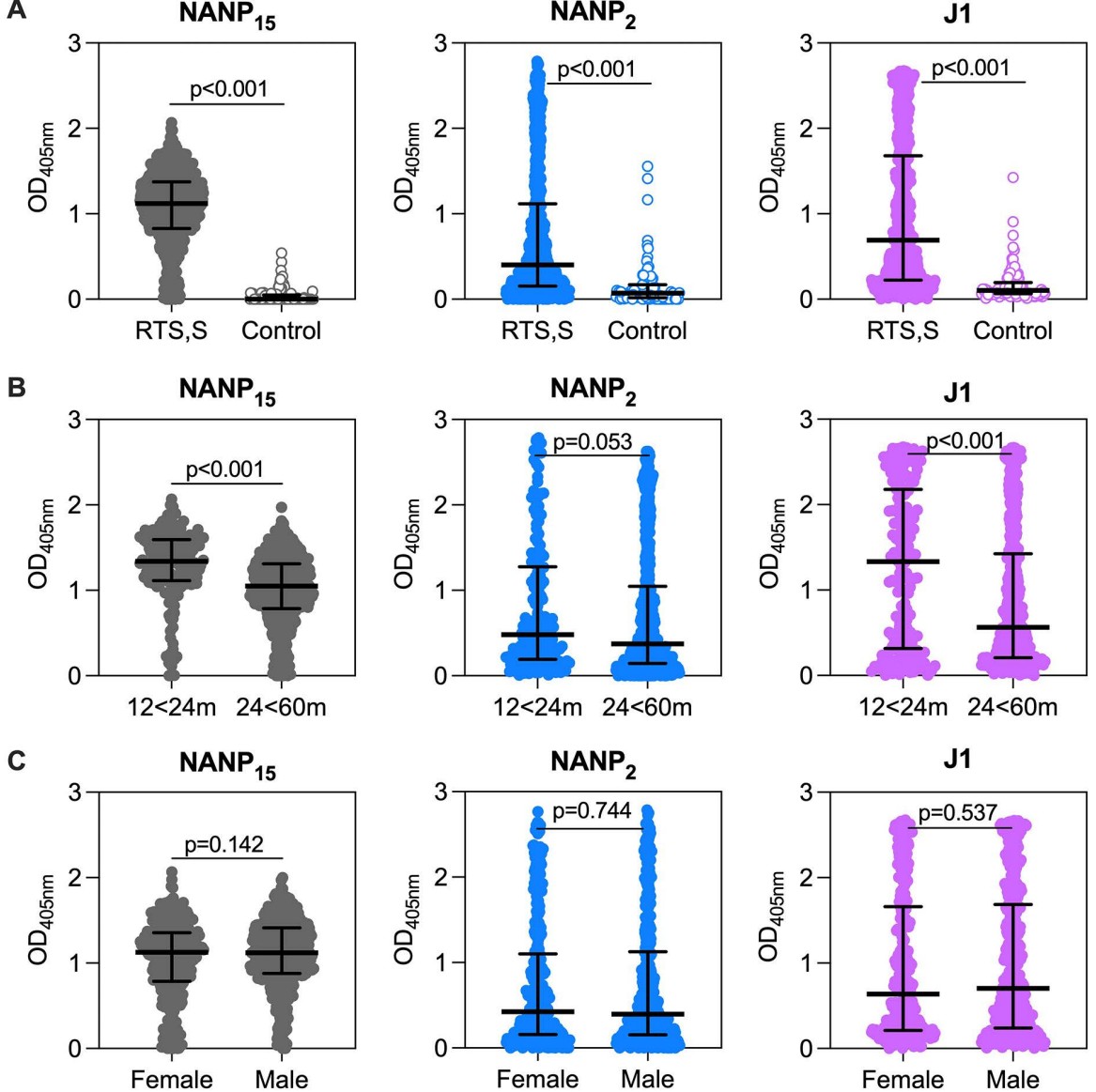

**Fig 3. IgG to CSP major repeat and junction epitopes in RTS,S vaccinated children.** Serum samples were collected from children who received RTS,S ($n=735$) or a control non-malaria vaccine ($n=99$) in a phase IIb clinical trial (study month 3 time point). **A)** Sera were tested in single for IgG to NANP$_{15}$ (1/1000 dilution), NANP$_2$ (1/500 dilution), and J1 (1/500 dilution) peptides by high-throughput ELISA; results show children who received the RTS,S vaccine (closed circle) vs. the control group (open circle) who received a non-malaria vaccine. RTS,S vaccinated children were sub-divided by **B)** age groups: 12<24 months ($n=176$; average, 17.7 months; range,12.3–23.9 months) or 24 < 60 months ($n=559$; average, 41.6 months; range, 24.0–59.8 months) and **C)** sex: female ($n=339$) or male ($n=396$). The Mann–Whitney $U$ test was performed to compare between groups. The OD median (centre line) and interquartile range (whiskers) of each group is shown.

vaccination ($n=646$; 29% had at least one episode; Fig 4A). Children were categorised as high and low responders based on the median IgG response (NANP$_{15}$, OD = 1.121; NANP$_2$, OD = 0.402; J1, OD = 0.691). Kaplan–Meier survival curves showed no significant difference in the time to the first malaria episode between high and low responders to the long NANP$_{15}$ peptide ($p=0.925$; Fig 4A), which is consistent with original findings reported from the phase IIb trial [30,36].

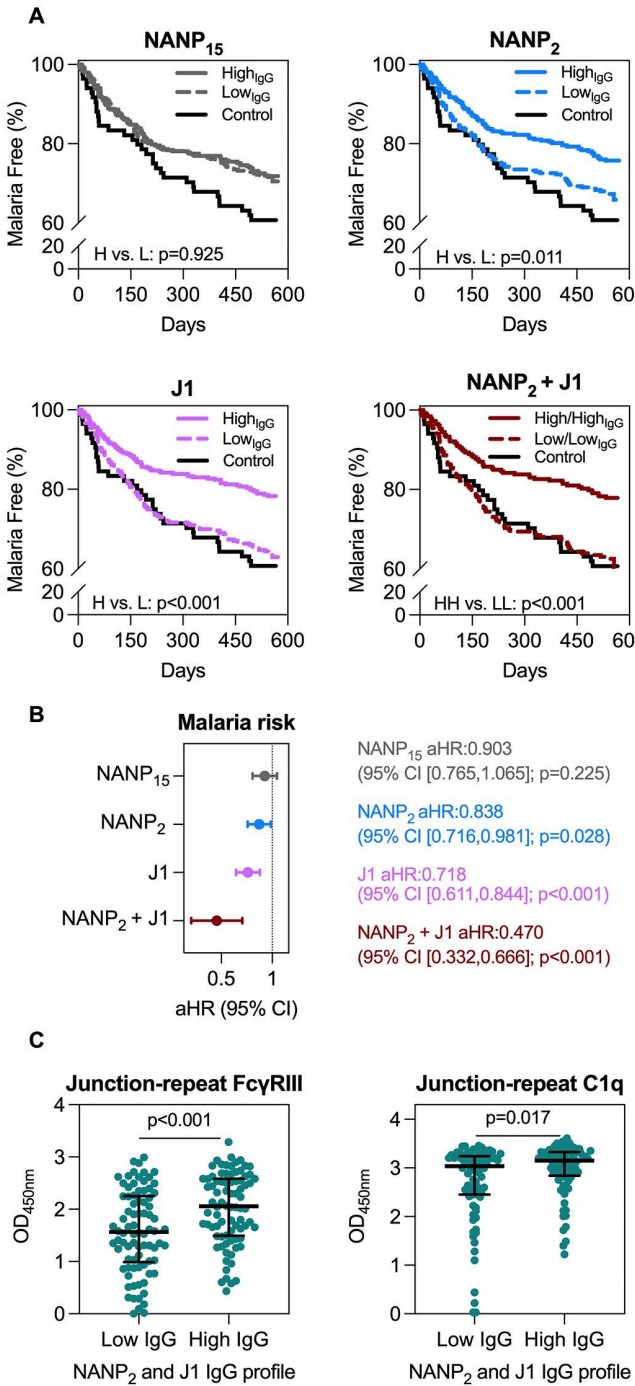

**Fig 4. Associations between IgG fine specificity and risk of malaria in RTS,S vaccinated children.** RTS,S vaccinated children were tested for IgG to NANP$_{15}$, NANP$_2$, and J1 peptides by ELISA and those with complete follow up data were used for analysis ($n = 646$ for RTS,S vaccine recipients and $n = 84$ for the control group). **A)** The time to first malaria episode was evaluated as Kaplan–Meier survival curves for high and low IgG responders to each peptide defined by the median response (NANP$_{15}$: OD = 1.121, high $n = 344$ and low $n = 302$; NANP$_2$: OD = 0.402, high $n = 339$ and low $n = 307$; J1: OD = 0.691, high $n = 342$ and low $n = 304$) measured at study month 3 time point. The Log-rank (Mantel-Cox) test was used to compare the high and low responders and corresponding $p$-values are shown. Children in the non-malaria control vaccine group were shown for comparison ($n = 84$). Of the 646 children with clinical data available, a subset of children had high IgG to both NANP$_2$ and J1 peptides (NANP$_2$+J1, $n = 254$) and we compared responses to children with low IgG to both NANP$_2$ and J1 peptides ($n = 220$). All children were present from day 0 through to the end of the study. **B)**

The Cox proportional-hazards model was performed to evaluate the adjusted hazard ratio (aHR; adjusted for age) with 95% confidence interval (CI) and corresponding $p$-values for a unit increase in IgG (normalised by the standard deviation) and malaria risk. C) Functional antibody responses to the Junction-repeat peptide were compared among children with high IgG to both J1 and $NANP_2$ epitopes ($n = 82$) or low IgG to both epitopes ($n = 82$). FcγRIII-binding and complement (C1q) fixation activity of antibodies is shown. The Mann–Whitney $U$ test was performed to compare between groups and corresponding $p$-values are shown. The OD median (centre line) and interquartile range (whiskers) of each group is shown.

However, children with high IgG to $NANP_2$ or J1 peptides had a significantly lower risk of malaria than those with low IgG ($p = 0.011$ and $p < 0.001$, respectively). Of the 646 children with clinical data available, a subset of children had high IgG to both $NANP_2$ and J1 peptides ($n = 254$); among these, there was a significantly reduced time to first malaria episode compared to children with low IgG to both $NANP_2$ and J1 ($n = 220$, $p < 0.001$). It was notable that children who had low reactivity to $NANP_2$ and J1 peptides had a malaria rate that was very similar to the control vaccine group. This suggests that IgG to J1 and $NANP_2$ effectively discriminate between children with significant vaccine immunity and those without. We used Cox-proportional hazards models to determine the adjusted hazard ratio (aHR; adjusted for age) for first malaria event for each unit increase in IgG (Fig 4B, Table B in S1 Appendix,). IgG to $NANP_2$ (aHR: 0.838 (95% CI [0.716, 0.981]; $p = 0.028$)) and J1 (aHR: 0.718 (95% CI [0.611, 0.844]; $p < 0.001$)) peptides were associated with a significantly reduced risk of clinical malaria. Furthermore, children who were high responders to both $NANP_2$ and J1 peptides had the strongest association with protection (aHR: 0.470 (95% CI [0.332, 0.666]; $p < 0.001$)). In contrast, there was no significant association for IgG to $NANP_{15}$ (aHR: 0.903 (95% CI [0.765, 1.065]; $p = 0.225$)). Therefore, IgG fine specificity to $NANP_2$ and J1 epitopes, but not IgG reactivity to the longer $NANP_{15}$ epitope, were significantly associated with clinical protection in young children living in a malaria endemic region.

To further understand the protective associations observed for IgG to the $NANP_2$ and J1 epitopes, we explored whether antibody fine specificity was associated with antibody functional activity in a subset of children for which we had sera available ($n = 461$ RTS,S vaccinated and $n = 23$ control group) (Fig 4C, Fig E in S1 Appendix). We synthesised a peptide that combined the junction, minor repeat, and major repeat sequences of CSP (referred to as Junction-repeat) that was suitable for use in established functional immunoassays. We quantified antibody-mediated FcγRIII-binding and C1q complement fixation, which are established mechanisms of protective immunity against sporozoites [36]; binding of C1q by antibodies is the first essential step in activation of the classical complement pathway. Of the 461 RTS,S-vaccinated children, a subset had high IgG to both J1 and $NANP_2$ ($n = 82$; Fig 4C) and among this group, there was significantly higher FcγRIII binding activity ($p < 0.001$) and complement fixation activity ($p = 0.017$) compared to the group of children with low IgG to both J1 and $NANP_2$ ($n = 82$).

## Variable antibody response profiles amongst RTS,S vaccinated children

We investigated the relationship between IgG responses to $NANP_{15}$, $NANP_2$ and J1 in RTS,S vaccinated children. IgG to $NANP_{15}$, $NANP_2$ and J1 peptides were only moderately correlated with each other (rho = 0.404–0.634, $p < 0.001$ for all tests; Fig 5A). We used hierarchical k-means clustering on the three antibody parameters to group children into four clusters, which were then evaluated using Principal Components Analysis (Fig 5B). The largest, cluster 3 ($n = 300$), was primarily characterised by having high IgG to $NANP_{15}$, while cluster 2 ($n = 139$) had high IgG to $NANP_{15}$ and J1, and cluster 1 ($n = 138$) had high IgG to all three peptides (Fig 5C). In contrast, cluster 4 ($n = 158$) included children who had low IgG to all peptides. These analyses further highlight that high IgG responses to J1 and $NANP_2$ epitopes, which were associated with protection, is restricted to a subset of children. Of interest, cluster 1 (high IgG to $NANP_{15}$, $NANP_2$ and J1 IgG) had the highest median FcγRIII-binding and C1q complement-fixation activity followed by cluster 2 (high IgG to $NANP_{15}$ and J1 IgG only), and cluster 3 (high IgG to $NANP_{15}$ IgG only; Fig 5D). Cluster 4 had low IgG responses to all epitopes and low functional activity and represents a sub-group of children with low responses, or hypo-responders. The presence of a substantial group of low responders may contribute to the moderate overall vaccine efficacy in this trial.

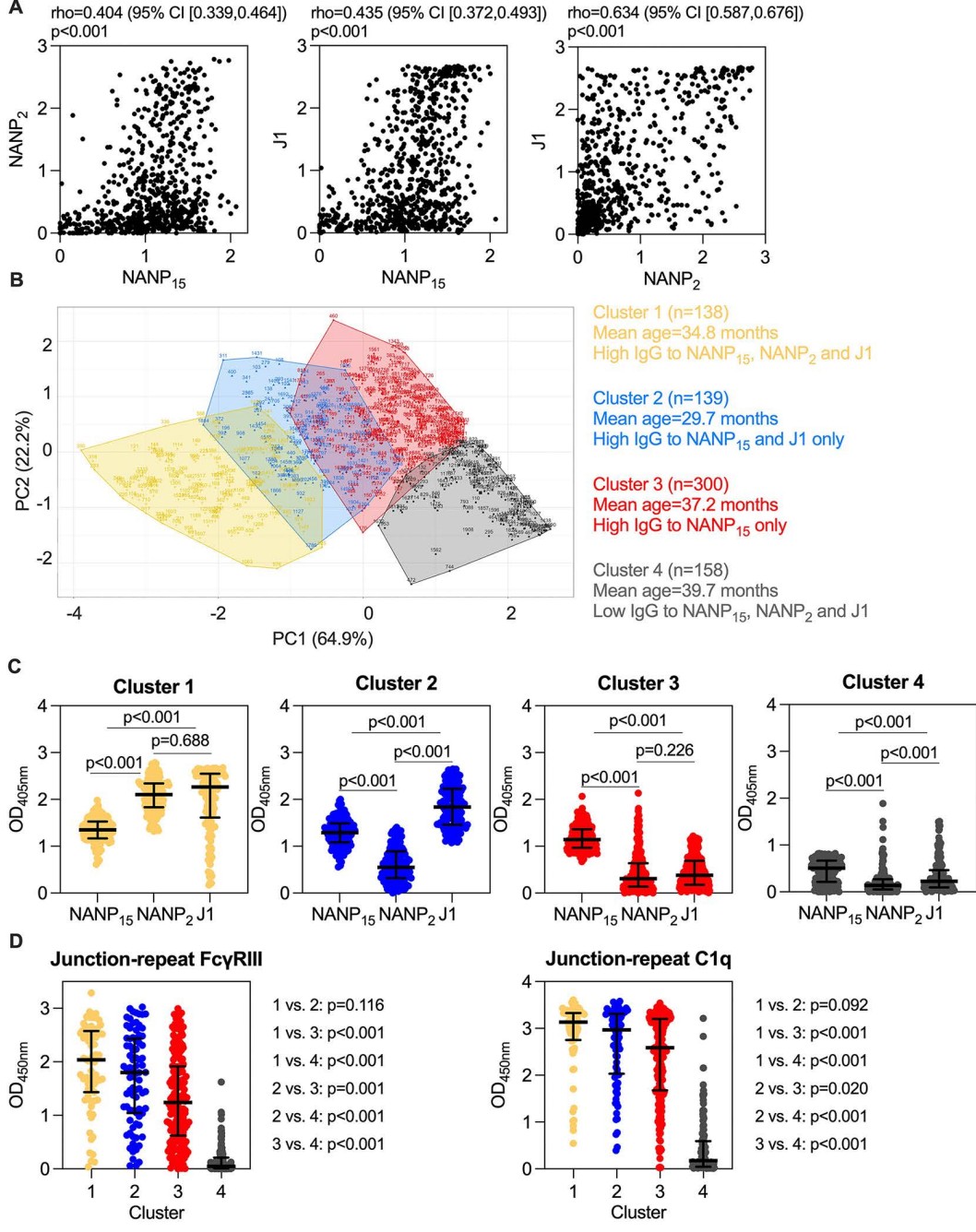

**Fig 5. Analysis of NANP$_{15}$, NANP$_2$ and J1 IgG response in RTS,S immunised children.** RTS,S vaccinated children ($n = 735$) were tested for IgG to NANP$_{15}$, NANP$_2$ and J1 peptides (at study month 3 time point) and **A)** shown on a scatterplot. The Spearman's correlation coefficient (rho) is shown with 95% confidence interval (CI) and corresponding *p*-values. **B)** Hierarchical k-means clustering was performed using the three antibody parameters to group children into four clusters, which were then evaluated using Principal Components Analysis, shown in the figure. IgG data is presented by principal component (PC) scores PC1 (64.9% variance) and PC2 (22.2% variance). Participants were coloured according to cluster and shaded polygons around cluster members with cluster centre dot were added. Four major clusters were identified: cluster 1 $n = 138$ (19% of samples), cluster 2 $n = 139$ (19% of samples), cluster 3 $n = 300$ (41% of samples) and cluster 4 $n = 158$ (21% of samples). **C)** The respective NANP$_{15}$, NANP$_2$ and J1 IgG for individuals within each identified cluster were plotted. The Wilcoxon signed-rank test was performed to compare between NANP$_{15}$, NANP$_2$ and J1 IgG responses of matched participants within each cluster and corresponding *p*-values are shown. **D)** Functional antibody responses (FcγRIII binding and C1q complement-fixation) to the Junction-repeat peptide were evaluated in a subset of children within each cluster (cluster 1 $n = 71$, cluster 2 $n = 81$, cluster 3

$n = 179$ and cluster 4 $n = 130$). The Mann–Whitney $U$ test was performed to compare between clusters and corresponding $p$-values are shown. The OD median (centre line) and interquartile range (whiskers) of each cluster is shown.

## Discussion

We evaluated the fine specificity of vaccine-induced IgG to short epitopes of the major NANP-repeat region as well as minor NVDP-repeat and junction region epitopes that are excluded from current malaria vaccines. In a large RTS,S phase IIb trial in Mozambique ($n = 735$ children), we found that a subset of children had vaccine-induced IgG with specificity to short NANP-repeats (NANP$_2$) and cross-reactivity to the junction epitope (J1: KQPADGNPDPNANPN). A major finding was that high IgG responses to these epitopes were associated with a significantly reduced risk of clinical malaria in young children aged 1–4 years, monitored for 18 months post vaccination. The protective association was strongest among children with high responses to both epitopes, NANP$_2$ and J1. It was notable that children with little or no IgG to NANP$_2$ and J1 had a rate of malaria similar to that of the control non-malaria vaccine group. Together, these findings indicate that quantifying IgG to these epitopes is able to differentiate between vaccinated children who were protected or non-protected against malaria. In contrast, IgG to longer NANP-repeat sequences, which is the current vaccine reference assay widely used for evaluating RTS,S and R21 vaccines, was not associated with protection in the same analysis. Having antibodies with epitope specificities to NANP$_2$ and J1 peptides was also associated with higher antibody Fc-mediated functional activities, which is relevant because IgG Fc-mediated functions have been linked with RTS,S protective immunity [36].

RTS,S vaccination in young African children and in preclinical mouse vaccine models could induce cross-reactive IgG that recognised NPDP and NVDP motifs excluded from the vaccine construct. The use of mouse vaccine models, which excludes the potential confounding of prior malaria exposure, was valuable for establishing that our assays could detect antibody specificity and cross-reactivity to our peptides. Interestingly, vaccination with full-length CSP, which contains the NPDP and NVDP motifs, did not appear to be superior to vaccination with the truncated RTS,S construct lacking these sequences. Our data from mouse models further suggests that the NANP-repeat sequence length in vaccine constructs may influence the extent of cross-reactivity of antibodies, with greater cross-reactivity observed with NANP$_9$ versus NANP$_4$ vaccine constructs. The observed cross-reactivity may be due to shared amino acids between NANP (Asparagine-Alanine-Asparagine-Proline), NVDP (Asparagine-Valine-Aspartic Acid-Proline) and NPDP (Asparagine-Proline-Aspartic Acid-Proline) motifs.

Pooled analysis of two smaller trials of malaria-naïve adults ($n = 61$ in total) given experimental RTS,S regimens also reported antibody binding to the junction sequence, which was associated with protection against homologous controlled human malaria infection performed 3 weeks post-vaccination [49,50]. However, in these trials RTS,S was co-administered with an adenovirus vaccine encoding CSP or a delayed fractional dosing regimen of RTS,S, neither of which are approved or recommended for children in Africa. Furthermore, protection against homologous experimental infection three weeks post-vaccination may not reflect vaccine immunity against clinical malaria episodes in children residing in endemic regions who are exposed to heterologous parasite strains. Antibody cross-reactivity to junction sequences and minor NVDP-repeats was recently reported in malaria-naïve adults immunised with R21, but has not been evaluated in relation to protective efficacy [51]. No prior study has evaluated antibody fine specificity or cross-reactivity in vaccinated children or in malaria endemic populations, nor have antibody specificities been linked with clinical protection against natural exposure to malaria and vaccine efficacy. The important finding from our study was that high IgG to NANP$_2$ and J1 epitopes correlated with vaccine efficacy against clinical malaria episode and poor responses to both epitopes could identify children with no protective immunity. Antibodies specific to J1 (KQPADGNPDPNANPN) may mediate protection by inhibiting the proteolytic cleavage of CSP which occurs in the neighbouring "Region I" (sequence KLKQP). This cleavage results in the CSP conformational changes required for hepatocyte attachment and entry [12,13]. This activity has been demonstrated

with CIS43, a mAb induced by whole sporozoite vaccination with specificity to PADGNPDPNANPNVD and NPDPNAN-PNVDPNAN epitopes similar in amino acid sequence to our J1 peptide [44]. The lack of protective associations for IgG to a longer NANP-repeat sequence in our study and others (summarised in S1 Appendix, Table A) is likely because assays using a long NANP-repeat sequence do not discriminate children with different antibody specificities or children with protective and non-protective responses, particularly since only a subset of children have antibodies with specificity for $NANP_2$ and/or the junction (J1) peptide that were associated with protection.

Our data further revealed that vaccination with RTS,S in children, or with different CSP-based constructs in mice, generated antibodies to NANP sequences of different lengths. Only a subset of children had substantial antibodies to the shortest $NANP_2$ epitope, whereas most children had IgG that recognised longer sequences; this has not been evaluated previously. High $NANP_2$ IgG responses were associated with protection from malaria, suggesting it may be a valuable correlate of protection, especially when combined with IgG to the J1 epitope. mAbs recognising short NANP sequences $((DPNANPNV)_2$, PNANPN, NPNANPNANPNA) that provided protection in mice against infection challenge have been previously reported [41,42,52]. The CSP central repeat region comprises multiple epitopes whereby antibodies bind to the region in a coiled conformation and therefore IgG specificity for an epitope of two NANP-repeats, rather than longer epitopes, would allow a greater number of IgG molecules to bind to the central repeat region [40,53]. This may enhance Fc-mediated functions that are implicated in protection, increase homotypic interactions between IgG molecules and enhance B cell activation to promote clonal selection and affinity maturation of human B-cells that express CSP protective antibodies [36,54]. In our analyses, higher Fc-mediated functional activities (C1q complement fixation and FcγRIII binding) were observed in children with high $NANP_2$ and J1 IgG profiles. Prior studies of functional antibodies in vaccine trials have only evaluated these using long NANP-repeat sequences and the C-terminal domain included in RTS,S and R21. Our findings suggest that future studies could further investigate the relative importance of antibodies targeting specific epitopes, or combinations, for functional activities.

By testing responses in a large cohort of vaccinated children, we were able to identify subgroups with different response profiles, and children could be classified into four main groups. Only a subset of children had high IgG to J1 and $NANP_2$, which were associated with protection in our longitudinal analysis. A further significant finding was the identification of children with low responses to all antigens, including $NANP_{15}$; these children may represent vaccine hypo-responders and understanding the basis for these poor responses is an important objective for future research [55,56]. Responses to the J1 epitope were notably lower in older compared to younger children. Related to this observation, efficacy of R21 was reportedly significantly lower in older children (18–36 months old) compared to young children (5–17 months), although these age groups are not directly comparable to the age of our study participants [4]. The basis for this age effect is unclear, but it could relate to cumulative malaria exposure prior to vaccination, other infections, nutritional parameters or other environmental factors that can impact immune function. Host genetic factors, such as HLA genotype, may also influence vaccine responses and efficacy. We did not find a strong effect on IgG responses to $NANP_{15}$, $NANP_2$ and J1 of malaria exposure, as reflected by IgG levels to *P. falciparum* merozoites. However, a prior analysis found that adjusting survival analyses based on antibody biomarkers of malaria exposure (IgG to *P. falciparum* erythrocyte membrane protein 1 and *P. falciparum* merozoite surface protein 2) did increase the protective associations found for vaccine-induced antibodies [57]. Currently, there is variability in the field with respect to antigens and assays used to evaluate vaccine-induced immunity with RTS,S and R21, and other CSP-based vaccine candidates. Future harmonisation of assays is needed, and our findings suggest that quantifying IgG to $NANP_2$ and junction peptides will be a valuable part of assessing vaccine responses in the future.

Given the strong protective associations observed in children with high IgG response to $NANP_2$ and J1 peptides, a potential strategy to improve vaccine efficacy may be to design CSP-based immunogens that induce stronger $NANP_2$ and junction (J1) IgG responses amongst all recipients. In our mouse studies, vaccination with full-length CSP did not appear to induce better junction responses than the RTS,S construct. Therefore, future work is needed to understand how

to best enhance vaccine responses to the junction and short NANP-repeat epitopes. Vaccine constructs including fewer NANP-repeats and/or incorporating the junction sequence could be considered. Some data suggest that antibodies that recognise both NPDP junction motifs and NANP-repeat motifs may provide better protection in animal models [41–43,58]. In mice, decreasing the number of NANP-repeats and including the junction amino acid sequence and minor NVDP-repeats has been shown to improve protection against challenge [59–61], in-line with our finding that antibodies to these regions correlate with clinical protection in children. However, other mouse studies [62,63] have not seen improved protection, making it difficult to draw conclusions from murine malaria challenge models alone as they may not well represent natural exposure to *P. falciparum* in human populations.

There are limitations to consider with this study. We identified two immunological correlates of protection for the RTS,S malaria vaccine in a large phase IIb cohort of Mozambican children. Given the similarities between the RTS,S and R21 vaccine constructs, our findings are likely relevant for both vaccines, but additional studies on R21 are needed in the future. Furthermore, it will be important to evaluate the correlates of protection identified here in other African populations with different malaria transmission intensities and determine how these responses are impacted after receiving a booster vaccine dose. While we have not demonstrated that the identified antibody specificity can confer protection against infection or disease in the animal vaccine models used, this may be valuable to assess in future studies. Lastly, we performed functional immunoassays using the Junction-repeat peptide comprising NANP, NVDP and junction NPDP motifs, but future work is required to optimise assays to detect functional antibodies to the short, biotinylated peptides specifically $NANP_2$ and J1.

In conclusion, we found that RTS,S vaccination in children induces IgG with specificity to the junction and short $NANP_2$ epitope, and these responses were associated with vaccine protection against malaria. There are two major implications of these findings. First, the data suggest that IgG to J1 and $NANP_2$ may be a valuable correlate of protection for RTS,S, which is needed to inform vaccine implementation and booster doses, vaccine surveillance, and monitoring populations at risk of malaria. Second, these findings may also inform the design of future more efficacious vaccines that can generate more potent and uniform IgG responses to the clinically important $NANP_2$ and J1 epitopes.

## Supporting information

**S1 Appendix. Supplementary methods. Table A.** Overview of published studies examining the association between IgG to NANP-repeats and protection from malaria among infants and young children residing in malaria-endemic countries immunised with the RTS,S vaccine. **Table B.** Cox proportional-hazard (CPH) model hazard ratios (HR) for $NANP_{15}$, $NANP_2$ and J1 IgG in RTS,S vaccinated children adjusted by age (aHR). **Fig A. Study design and statistical analysis approach. A)** Summary of the preclinical and clinical samples tested for IgG to six CSP-specific peptides, indicating what data are included in each figure. **B)** The causal framework for analysis to identify correlates of protection (outcome) in RTS,S vaccinated children (exposure) is demonstrated through a Directed Acyclic Graph (DAG). Age and sex were considered as potential confounders, a priori, but only age was associated with vaccine responses and was included in the final adjusted model. No other confounders were identified. **Fig B. IgG to $NANP_{15}$, $NANP_3$ and $NANP_2$ peptides in a subset of children vaccinated with RTS,S.** A subset of serum samples ($n = 40$, study month 3 time point) from children who received RTS,S in a phase IIb clinical trial were **A)** tested for IgG to $NANP_{15}$, $NANP_3$ and $NANP_2$ at a 1/500 dilution by ELISA. The OD median (centre line) and interquartile range (whiskers) of each group is shown. **B)** IgG data for each peptide is also shown on a scatterplot with the Spearman's correlation coefficient (rho) and 95% confidence interval (CI), corresponding p-values shown. **Fig C. IgG to $NANP_{15}$, $NANP_2$ and J1 peptides by *P. falciparum* exposure (merozoite IgG) in RTS,S vaccinated children.** Vaccinated children ($n = 715$) were tested for IgG binding to merozoites by ELISA as a biomarker of *P. falciparum* exposure. The IgG responses to merozoites were categorised as high ($n = 357$) and low ($n = 358$) exposure based on the group median OD = 0.220. The median of the high exposure group was OD = 0.442 (range, 0.220,1.206) and of the low exposure group was OD = 0.100 (range, 0.000,0.200). Vaccine IgG responses to

NANP$_{15}$, NANP$_2$ and J1 are shown for high versus low exposure groups. The OD median (centre line) and interquartile range (whiskers) of each group is shown. The Mann–Whitney $U$ test was performed to compare between groups and corresponding p-values are shown. **Fig D. NANP$_{15}$, NANP$_2$, and J1 IgG correlation with NANP$_{15}$ avidity.** Sera from a subset of RTS,S vaccinated children ($n = 36$, study month 3 time point) were tested for antibody avidity to NANP$_{15}$ IgG and the data was correlated with IgG to NANP$_{15}$, NANP$_2$, and J1 peptides and shown on a scatterplot. The Spearman's correlation coefficient (rho) is shown with 95% confidence interval (CI), corresponding $p$-values are shown. **Fig E. IgG and functional antibody responses to the Junction-repeat peptide comprising NANP, NPDP and NVDP motifs in a subset of children immunised with RTS,S or a control non-malaria vaccine.** Serum samples were collected from children who received three doses of RTS,S ($n = 461$) or a control non-malaria vaccine ($n = 23$). Sera were tested in duplicate for Junction-repeat-IgG at a 1/500 dilution factor, Junction-repeat-FcγRIII binding at a 1/500 dilution factor and Junction-repeat-C1q fixation at a 1/100 dilution factor by ELISA. The OD median (centre line) and interquartile range (whiskers) of each group is shown. The Mann–Whitney $U$ test was performed to compare between vaccine groups and corresponding $p$-values are shown.
(PDF)

**S1 Data. A de-identified database including antibody specificity data for preclinical and clinical samples.**
(XLSX)

**S1 Checklist. Inclusivity in global research form.**
(PDF)

**S2 Checklist. The Animal Research: Reporting of In Vivo Experiments (ARRIVE) guidelines 2.0: author checklist.**
(PDF)

**S3 Checklist. Strengthening the Reporting of Observational Studies in Epidemiology (STROBE) guidelines.** Information on the STROBE initiative is available at https://www.strobe-statement.org/, licensed under the Creative Commons Attribution 4.0 International (CC BY 4.0). The STROBE checklist originates from von Elm E et al. 2008. *J Clin Epidemiol, 61*(4) and can be accessed on https://doi.org/10.1016/j.jclinepi.2007.11.008.
(PDF)

## Acknowledgments

We thank all children, parents and guardians who participated in this study. We thank all investigators and staff at CISM (Mozambique) and ISGlobal (Spain) who were involved in previously conducting the clinical trial and collecting clinical and trial data, staff in the participating laboratories for coordinating sample management and shipment, and Didac Maciá and Luis Molinos-Albert for comments on the manuscript. Burnet Institute is located on the traditional lands of the Boonwurrung people of the Kulin Nations and we acknowledge their elders past and present.

## Author contributions

**Conceptualisation:** Alessia Hysa, D. Herbert Opi, Liriye Kurtovic, James G. Beeson.

**Data curation:** Alessia Hysa, Joshua Waterhouse, Liriye Kurtovic, James G. Beeson.

**Formal analysis:** Alessia Hysa, D. Herbert Opi, Joshua Waterhouse, Liriye Kurtovic, James G. Beeson.

**Funding acquisition:** D. Herbert Opi, Liriye Kurtovic, James G. Beeson.

**Investigation:** Alessia Hysa, Sandra Chishimba, Jessica L. Horton, Natalie Kingston, Hans J. Netter, David Wetzel, Michael Piontek, Gaoqian Feng, Liriye Kurtovic, James G. Beeson.

**Methodology:** Alessia Hysa, D. Herbert Opi, Sandra Chishimba, Jessica L. Horton, Natalie Kingston, Hans J. Netter, David Wetzel, Michael Piontek, Gaoqian Feng, Liriye Kurtovic, James G. Beeson.

**Project administration:** D. Herbert Opi, Liriye Kurtovic, James G. Beeson.

**Resources:** Jahit Sacarlal, Carlota Dobaño, Liriye Kurtovic, James G. Beeson.

**Software:** Liriye Kurtovic, James G. Beeson.

**Supervision:** D. Herbert Opi, Liriye Kurtovic, James G. Beeson.

**Validation:** Alessia Hysa, Liriye Kurtovic, James G. Beeson.

**Visualisation:** Alessia Hysa, Joshua Waterhouse, Liriye Kurtovic, James G. Beeson.

**Writing – original draft:** Alessia Hysa, D. Herbert Opi, Liriye Kurtovic, James G. Beeson.

**Writing – review & editing:** Alessia Hysa, D. Herbert Opi, Liriye Kurtovic, James G. Beeson.

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
