## [Editor Report · Decision Letter 0]

15 Dec 2025

Dear Dr Beeson,

Thank you for submitting your manuscript entitled "Antibody fine specificity reveals new correlates of protection for the RTS,S malaria vaccine in young African children." for consideration by PLOS Medicine.

Your manuscript has now been evaluated by the PLOS Medicine editorial staff, and I am writing to let you know that we would like to send your submission out for external peer review.

Please also add a paragraph on study limitations in the Discussion, and also discuss whether these results can be validated in additional samples from other RTS, S and/or R21 trials. Please confirm that the use of the samples in this study is covered by the original consent agreement. Please also acknowledge that you have not demonstrated that the identified antibody specificity can confer protection against infection or disease in animal models.

For clinical studies, please upload a copy of your trial study protocol as a supporting information file. The study protocol should be the version submitted for approval to the institutional review board or ethics committee, should include any amendments to the study protocol, as well as the date of their approval by the institutional review or ethics committee. Please also detail any deviations from the study protocol in the Methods section of your manuscript. The editors will consider the protocol and study conduct prior to a final decision for external review.

Please re-submit your manuscript within two working days, i.e. by Dec 17 2025 11:59PM.

Kind regards,

Alison Farrell, Ph.D.

Senior Editor

PLOS Medicine

---

## [Decision Letter · Decision Letter 1]

23 Jan 2026

Dear Dr Beeson,

Many thanks for submitting your manuscript "Antibody fine specificity reveals new correlates of protection for the RTS,S malaria vaccine in young African children." (PMEDICINE-D-25-04372R1) to PLOS Medicine. The paper has been reviewed by subject experts and a statistician; their comments are included below and can also be accessed here: [LINK]

As you will see, the reviewers find the results interesting and provocative, but raise concerns regarding the lack of harmonization of the data from humans and mice, missing information and the need for justification of some of the methodological choices. Reviewer 1 also questions how samples were handled, potentially affecting some of the results, and whether ethical approval extends to the current analyses. After discussing the paper with the editorial team and an academic editor with relevant expertise, I'm pleased to invite you to revise the paper in response to the reviewers' comments. We plan to send the revised paper to some or all of the original reviewers, and we cannot provide any guarantees at this stage regarding publication.

We ask that you submit your revision by Feb 13 2026 11:59PM. However, if this deadline is not feasible, please contact me by email, and we can discuss a suitable alternative.

Don't hesitate to contact me directly with any questions (afarrell@plos.org).

Best regards,

Alison

Alison Farrell, Ph.D.

Senior Editor

PLOS Medicine

afarrell@plos.org

Comments from the reviewers:

Reviewer #1: The development and evaluation of malaria vaccine based on an immune response to the CSP protein has been hindered by the lack of a reliable serological correlate of protection. Thus, the findings of this paper that antibody titres to a short NANP repeat and to a peptide at the junctional region of the CSP antigen provide a stronger correlation of protection than the ELISA used in most previous studies is important and could aid future vaccine development based on the CSP antigen, including RNA vaccine. This is an important finding. The investigators describe progression through a logical series of approaches to demonstration of the association between vaccine induced responses to these antigen and protection against clinical episode of malaria in young African children. The paper is well written, and results including the figures presented clearly so I have only a few comments and suggestions.

Methods

Monoclonal antibodies: It is not clear why the specific monoclonal antibodies tested were selected for these studies. Could this be explained?

Sera from the phase 2 trial in Mozambique: These samples are now 20 years old and I wonder if they have been thawed and frozen on previous occasions for other studies. Were original aliquots used?.

Statistical analysis: Was a formal SAP developed before the results from the Mozambiquan study were analysed?

Ethics: Ethics for the clinical trial was obtained from the appropriate ethics committees at the time of the study. Did the consent form include provision for retention of the samples for this long period and for their use in subsequent laboratory studies?

Results

I am not qualified to comment on the details of the immunological assays that were used but these are primarily assays that have been well validated previously and conducted by the authors. Results are presented clearly and were easily followed including the figures.

Murine studies: The numbers of mice included in the different groups was generally very small for this kind of study., Was there a reason for this?

Combination of J1 and NANP2: Was the combination associated with a significantly greater level of protection against clinical malaria than either peptide given alone.

Discussion

Full length peptide: Do the investigators have an explanation of why this long peptide does not provide a more effective immune response?

Heterogeneity in response: A striking feature of many serological studies conducted in African children vaccinated with RTS,S or R21, as in this study, is the wide range of response in the anti-CSP antibody response to the vaccine. Age was shown to be a factor in this study, with older children doing less well than younger ones, as observed in the R21 phase 3 trial, but gender and overall malaria exposure were not. Is this heterogeneity likely to involve genetic factors? Do the authors have any other ideas as to what could account for this heterogeneity and are they undertaking any further studies to investigate this?

Reviewer #2: This was a nicely written paper with interesting findings.

It would be good to include the transmission rates/seasonality in Mozambique (and the specific area the study was conducted) at the time of the study.

Overall, the rationale for the use of 2,3 and 15 repeats of NANP needs a more robust justification, especially when many of the papers cited (and many in the field use) NANP with 6 repeats. Do the authors have data on IgG responses with NANP6? This would be an important addition.

It;s important to describe the reasons why mice were vaccinated with NANP4, NANP9 and NANP19 (RTS,S), but ELISAs developed for NANP2, NANP3 and NANP15. Why the difference?

The timepoints of clinical samples used is not speficied and is needed.

More information and justification is needed for the choice of cinical samples for various assays (not consistent between assays).

The use of the word "validation" for assays in clinical immunology is quite specific and I don;t think that the use of mAbs here satisfies what is generally considered to be an assay validation (https://www.ema.europa.eu/en/ich-q2r2-validation-analytical-procedures-scientific-guideline). Suggest re-wording.

Please add justification for running functional assays using an amino acid sequence that is not used in ELISAs (the data for which is used as justification for running the functional assays). Additionally, the functional assay references cited do not use the same sequence as used here. Please be more transparent about the differences and possible consequences of using different amino acid sequences for antigens in previous work by the group (and in references cited) and in the different quantification vs functionality work in this paper. This would be a good opportunity to highlight that the field in general is not very aligned and there is a pressing need for harmonisation (given the finding here of NANP2 (but not 3 or 15) being associated with vaccine-induced protection). For the functional assays, it would be interesting to see these responses to either a longer NANP construct or an otherwise lengthened/stabilised (perhaps with BSA?) J1 epitope to overcome the issue of peptide length. The solution here of using a compound peptide is that it's not known which epitope the functional response is directed to.

The papers cited on Fc-mediated activity of post-immunisation IgG (refs 43,44,45), seem like odd choices. Neither of the Behet papers measure FcR binding, the 2018 one looks at complement deposition, and the 2014 looks at inhibition of hepatocyte traversal but both using whole sporozoites incubation with post-immunisation IgG.

Were there any genetic differences between individuals who did not get any episodes of malaria? Was this investigated as part of this study?

Specific line items:

Line 41: Should use updated 2024 stats

Line 69: 19 repeats in RTS,S but 18 in R21

Line 138 - were all available samples representative of the whole? Why were only this number available?

What was the rate of malaria in the unvaccinated group?

Please add compound peptide junction sequence used for functional assays into table 1

-Line 169: They used a parametric stats test for preclinical results and non-parametric for clinical—is this conventional?

Line 173 "low and high responders" based in what? Which antigen?

Line 198: The mice were vaccinated using a homemade 'RTS,S' which is different to Mosquirix. How reliable is this?

Line 204 - add detail on seropositivity analysis

Line 207 - were the proportions all overlapping?

Line 213 why did they decide to vaccinate with nanp4 and nanp7 but test Ab with nanp2,3 and 15?

Line 214 says nanp9 whereas line 213 says 7. What is "significant" igg?

Line 227 "at higher concentrations...." - do they actually have concentrations? Than what?

Line 231 " all, including... except" reword

Line 234/5 "validated" suggest reword based on above comment.

line 243 samples chosen for pool n=50 - what timepoints? How many malaria episodes (in what time period) did the vaccinated people have? How many malaria episodes (in what time period) did thecontrol vaccinated people have?

line 245 what does "detectable" mean?

line 248 - include p values

line 250 - was avidity also investigated for NANP2, J1 as well as NANP15? If not, why not?

Line 256 (and Fig 3 B) - please include mean/range of ages within age groups 12-24 and 24<60 months

line 257 "both tests" should this be "both antigens"?

line 261 - correlation analysis - why was avidity only investigated for NANP15 and were these adjusted for multiple comparisons? Authors use 'avidity' and 'affinity' to mean the same thing here. I think it should be avidity both times.

line 269 - categorisation of "high" and "low" - based on what? At which timepoint?

line 277 - please include the time to first episode for each group

line 280 - what was the control vaccine?

line 318 - suggest move to discussion

line 327 - add ages of "young" children

line 332 - suggest to amend "effectively" to "is able to differentiate" since this has not been used in a validation cohort

line 337 - given tihs paper relies upon the difference between length of sequence, please include some comment on the sequence used for functional assays and comparability to previous work showing complement and FcyR3 binding.

Line 338: The peptide sequence used to analyse functional antibody responses has not been used before. The references cited to support these functional activities as protective responses all use different antigens in their assays:

o Ref 26 (same as ref 43): Kurtovic et al 2024, use (NANP)15 and some of the C-term (separate assays)

o Ref 44: Behet et al 2018, use whole sporozoite assay—here they don't actually assess FcyR binding, they look at complement deposition and hepatocyte invasion.

o Ref 45: Behet et al 2014, also a whole sporozoite assay. Here they don't measure complement deposition or FcR binding, just the activity of IgG elicited by whole sporozoite immunization to inhibit hepatocyte traversal in vitro.

line 361 - amend "was" to "has not been evaluated"

Line 367: It would be interesting to use these outcomes in a predictive model to see if this statements 'identify children with no protective immunity' is reproducible.

line 395 - please comment on the fact that the most discriminitory group (high NANP2 and J1 responders) did not come out as a separate grouping. "longitudinal" - is this a longitudinal analysis? Do you mean longitudinal serological analysis or length of follow up for malaria episodes. This is confusing.

line 400-402 - ages groups are NOT the same between RTSS vaccinated children in this study and R21 - please highlight this.

line 408-409 - which markers of malaria exposure were used in ref 58?

Figures/Tables

Fig1 - please include the sequence use din functional assays

Fig 1 legend line 680 - "RTSS-like construct" would be good to reference work to evidence this

Fig 2 - please add pvalues in each plot and specify timepoints samples were

Fig 3 B - age groups not comparable to R21 - cross ref line 400-402 in discussion

Fig 3 B - please include mean/range of ages within age groups 12-24 and 24<60 months

Fig 4 - why are there only n=84 "controls"? How were they chosen? Are they representative of all controls? Line 251 says there were n=99. Again, how were the n=79 chosen for "high" and "low"? Line 276 says n=256 for "high" and n=220 for "low"

Fig 5B - please include the description for the clusters in the figure "High IgG to all 3", "high IgG to NANP15 and J1" etc. Fig 5D - please include n's for this.

supplementary methods

Why were (and what could be the consquences for this work) of using 3 different strains of mice?

PLease include brief details of antigen manufacture

Please include brief details of merozoites

Please include catalogue details of reagents used in all assays, including functionality.

supplementary figures

Figure S1 - please include details of merozoite ODs (mean/SD/range) by exposure group

Figure S2 - WHy are authors only showing correlations between NANP15 avidity and NANP2 and J1 IgG? Where is NANP15 IGG and NANP15 avidity? Why are data on avidity to NANP2 and J1 not shown? Suggest to have a heatmap of all correlations and then adjust for multiple comparisons.

Figure S3 - it is not clear from the legend if this is the same study or not. The samples numbers are very different (n=461 and n=23 - how were these chosen if this is the same study?) What is the "non-malaria comparator vaccine" used here? Is this showing IgG responses to the "junction repeat" sequence that is used in the functional assays? If so, why is this not shown in a main figure? And why was it not done on all samples?

Reviewer #3: This is a mixed method study that used both animal model and clinical participants from a randomised controlled trial study. I am not a animal model expert and only reviewed the statistical method for that part. The study design from the RCT appears appropriate. I have several questions about the statistical methods:

1. Please specify the rationale for using two-sample t-test in the animal model but the non-parametric Mann-Whitney U test for human participants.

2. Correlation coefficient was used to examine the relationship between IgG specificity to different peptides - this assumes linear dependence. Please specify if this assumption was examined. The scatterplot shown in Fig 5 is so noisy that it isn't clear if linear assumptions were appropriate.

3. Line 175 - shouldn't it be log rank *test*.

4. Only age was adjusted in the Cox model - was only age thought to be the only confounder / prognostic factor? Were there any clinical factors that could affect antibody response as well as malaria risk? Please also specify if proportional hazard assumption holds.

5. Line 181 - 'cluster analysis' instead of 'clustering'?

Reviewer #4: Hysa et al report on the important finding of antibody fine specificity to minimal epitopes as correlates for the RTS,S malaria vaccine in young children. The strengths of the manuscript include the important area of study, unique clinical cohort, and experimental approach to assess binding specificity. Overall this study reports novel information on antibody specificity and subgroup analyses to better understand antibody correlates in young children. However, the correlates analyses lacks a pre-specified analytical plan, the link between the preclinical and pooled data with he clinical trial is not clear; and the overall analysis lacked information on other immune features, demographics or other confounding variables.

Specific comments:

It is not clear how the inclusion of the preclinical model informs the evaluation of the presented evaluation of the human trial. It is noted that the mouse vaccination study 'validated antibody specificity and cross-reactivity'; yet it is not clear how this was done given the substantial differences between murine and human antibodies.

The text mentions validated assays were utilized, but information on the validation (repeatability, robustness, linearity etc) is not provided.

For the statistical and CoP analyses, is there a threshold of Ab specificities that are critical?

---

* Please upload any figures associated with your paper as individual TIF or EPS files with 300dpi resolution at resubmission; please read our figure guidelines for more information on our requirements: http://journals.plos.org/plosmedicine/s/figures. While revising your submission, we strongly recommend that you use PLOS's NAAS tool (https://ngplosjournals.pagemajik.ai/artanalysis) to test your figure files. NAAS can convert your figure files to the TIFF file type and meet basic requirements (such as print size, resolution), or provide you with a report on issues that do not meet our requirements and that NAAS cannot fix.

After uploading your figures to PLOS's NAAS tool - https://ngplosjournals.pagemajik.ai/artanalysis, NAAS will process the files provided and display the results in the "Uploaded Files" section of the page as the processing is complete.

If the uploaded figures meet our requirements (or NAAS is able to fix the files to meet our requirements), the figure will be marked as "fixed" above. If NAAS is unable to fix the files, a red "failed" label will appear above.

When NAAS has confirmed that the figure files meet our requirements, please download the file via the download option, and include these NAAS processed figure files when submitting your revised manuscript.

* Please ensure that the study is reported according to the appropriate guidelines and include the completed checklists as Supporting Information. When completing the checklist, please use section and paragraph numbers, rather than page numbers. Please add the following statement, or similar, to the Methods: "This study is reported as per [XXXX] guideline (S1 Checklist)."

FIGURES AND TABLES

SUPPLEMENTARY MATERIAL

REFERENCES

ANIMAL STUDIES

* Please report your study according to the ARRIVE guidelines and checklist http://www.nc3rs.org.uk/arrive-guidelines In the checklist please include sufficient text excerpted from the manuscript to explain how you accomplished all applicable items.

OBSERVATIONAL STUDIES

* Abstract: Please include the study design, population and setting, number of participants, years during which the study took place (enrollment and follow up), length of follow up, and main outcome measures.

* Please ensure that the study is reported according to the STROBE (or appropriate STOBE extension) guideline (available from: https://www.equator-network.org/reporting-guidelines/strobe) and include the completed STROBE (or STROBE extension) checklist as Supporting Information. Please add the following statement, or similar, to the Methods: "This study is reported as per the Strengthening the Reporting of Observational Studies in Epidemiology (STROBE) guideline (S1 Checklist)." When completing the checklist, please use section and paragraph numbers, rather than page numbers.

* For all observational studies, in the manuscript text, please indicate: (1) the specific hypotheses you intended to test, (2) the analytical methods by which you planned to test them, (3) the analyses you actually performed, and (4) when reported analyses differ from those that were planned, transparent explanations for differences that affect the reliability of the study's results. If a reported analysis was performed based on an interesting but unanticipated pattern in the data, please be clear that the analysis was data driven.

* Please state in the Methods section whether the study had a prospective protocol or analysis plan. If a prospective analysis plan (from your funding proposal, IRB or other ethics committee submission, study protocol, or other planning document written before analyzing the data) was used in designing the study, please include the relevant document(s) with your revised manuscript as a Supporting Information file to be published alongside your study and cite it in the Methods section. A legend for this file should be included at the end of your manuscript. If no such document exists, please make sure that the Methods section transparently describes when analyses were planned, and when/why any data-driven changes to analyses took place. Changes in the analysis, including those made in response to peer review comments, should be identified as such in the Methods section of the paper, with rationale.

---

## [Decision Letter · Decision Letter 2]

6 Apr 2026

Dear Dr. Beeson,

Thank you very much for re-submitting your manuscript "Antibody fine specificity reveals new correlates of protection for the RTS,S malaria vaccine in young African children." (PMEDICINE-D-25-04372R2) for review by PLOS Medicine.

I have discussed the paper with my colleagues and the academic editor and it was also seen again by three reviewers. I am pleased to say that provided the remaining reviewer, editorial and production issues are dealt with we are planning to accept the paper for publication in the journal.

Please include a response to the remaining concerns of the statistical reviewer and revise the manuscript to acknowledge the limitations.

[LINK]

We look forward to receiving the revised manuscript by Apr 13 2026 11:59PM.

Sincerely,

Alison Farrell, Ph.D.

Senior Editor

PLOS Medicine

plosmedicine.org

Requests from Editors: Please provide a rebuttal clarifying how you have addressed these points.

Please review your text for claims of novelty or primacy (e.g. 'for the first time') and remove this language. In addition, please check that any use of statistical terms (such as trend or significant) are supported by the data, and if not please remove them.

Please provide URLs for all funders and provide grant numbers for all relevant instances.

Please revise the Data Availability Statement to remove the 'reasonable request' phrasing and specify all requirements or restrictions to data access.

Please use RTS,S/AS01 at first mention of the vaccine in the Abstract and follow it with (RTS,S).

Similarly, please use the full name of R21 at first mention in the Introduction.

Line 30, revise "Vaccine-induced CSP mechanisms of immunity and correlates of protection are not well defined..." to "Mechanisms of immunity and correlates of protection for RTS, S are not well defined..."

Line 36: "We evaluated responses" should be "We evaluated antibody responses".

Please also move ' monoclonal antibodies ' to the end of the sentence and add their specificity. E.g.

"(n=735), as well as CSP-specific monoclonal antibodies."

If the clinical trial is named, please add the name.

Please start the paragraph on line 38 with a one sentence discussion of the results from the mouse studies and analyses of mAbs. As written, the Abstract does not mention the findings in the preclinical studies or rationale for their inclusion.

Line 38: Please refer to the name of the clinical trial here or refer to it as the phase 2 trial and revise as "In analyzing antibody responses in the phase 2 trial, we found..."

Please clarify in the Abstract that you performed a post hoc analysis of clinical trial samples.

* Please include statistics in the Abstract and more specifics about the key findings.

* Please use the active voice throughout.

* The Author Summary also makes no mention of the mouse studies. Please add a bullet point pertaining to them, explaining why they are included in this study.

* Please temper the conclusions in the Author Summary, final bullet point. The study identifies candidate correlates of protection. Please reframe accordingly. Please explain that more studies are needed to independently validate and confirm the association and to test whether eliciting these antibodies would confer protection against malaria.

* In the author summary, in the final bullet point of 'What Do These Findings Mean?', please include the main limitations of the study in non-technical language.

The statements regarding the clinical trial in the Acknowledgments should be moved to a separate Disclaimer subheading (lines 864-867).

In Fig 5c, please confirm that the P values comparing NANP15 and J1 are correct in all 4 clusters.

Please add a statement to the Methods that you have not corrected for multiple testing.

* Authors DW and MP are employees of ARTES Biotechnology Gmbh. Please therefore revise the competing interests statement.

* Please also add this statement to the manuscript's Competing Interests: "[Initials] is an Academic Editor on PLOS Medicine's editorial board."

* Please revise your title to comply with PLOS Medicine's style. Your title must be nondeclarative and not a question. It should begin with main concept if possible. "Effect of" should be used only if causality can be inferred, i.e., for an RCT. Please place the study design ("A randomized controlled trial," "A retrospective study," "A modelling study," etc.) in the subtitle (ie, after a colon).

* Please ensure that the Introduction ends with a clear description of the study question or hypothesis.

* Please ensure that all abbreviations are defined at first use throughout the text.

* Please confirm that all numbers presented in the abstract are present and identical to numbers presented in the main manuscript text.

Line 266: please suggest a possible reason

Lines 297, 300: Are the children in this trial from a single country? If so, please identify it.

Line 306, please correct grammar.

Line 310: Please qualify explore (i.e. explore what?)

Line 310: Please refer to testing samples from children, not to testing children, as written

Line 428: Please revise 'which were' to 'which was'

Line 498-500: sentence uses valuable twice. Please revise.

"* Statistical reporting: Please revise throughout the manuscript, including tables and figures.

- Please report statistical information as follows to improve clarity for the reader ""22% (95% CI [13,28]; p</=)"".

- Please separate upper and lower bounds with commas instead of hyphens as the latter can be confused with reporting of negative values.

- Please repeat statistical definitions (HR, CI etc.) for each set of parentheses."

"* PLOS defines the “minimal data set” to consist of the data set used to reach the conclusions drawn in the manuscript with related metadata and methods, and any additional data required to replicate the reported study findings in their entirety. Authors do not need to submit their entire data set, or the raw data collected during an investigation. Please submit the following data:

The values behind the means, standard deviations and other measures reported;

The values used to build graphs;

The points extracted from images for analysis."

* Please define all elements of box plots in the figure caption - center line, box limits and whiskers.

* Please ensure that where relevant figures include 95% CIs.

* Please show graph axes beginning at zero. If this is not possible, please show a break in the axis.

* When a p value is given, please specify the statistical test used to determine it in the legend.

* Where data points are discrete, please ensure that they are depicted in the figures as discrete data and not as a continuous line.

* In the Kaplan-Meier curve(s) please provide the number at risk for each time interval.

*Please clearly state in the Methods why the peptides used in the mouse studies differ from those in the analysis of human samples. While you have responded to the referees about this issue, it is not sufficiently explained in the manuscript.

* Please report your study according to the ARRIVE guidelines and checklist http://www.nc3rs.org.uk/arrive-guidelines In the checklist please include sufficient text excerpted from the manuscript to explain how you accomplished all applicable items.

* Please also include a completed STROBE checklist. Please add the following statement, or similar, to the Methods: ""This study is reported as per the Strengthening the Reporting of Observational Studies in Epidemiology (STROBE) guideline (S1 Checklist).""

When completing the checklist, please use section and paragraph numbers, rather than page numbers."

"* Did your study have a prospective protocol or analysis plan? Please state this (either way) early in the Methods section.

c) In either case, changes in the analysis-- including those made in response to peer review comments-- should be identified as such in the Methods section of the paper, with rationale."

* Your study is observational and therefore causality cannot be inferred. Please remove language that implies causality and refer to associations instead.

* For all observational studies, in the manuscript text, please indicate: (1) the specific hypotheses you intended to test, (2) the analytical methods by which you planned to test them, (3) the analyses you actually performed, and (4) when reported analyses differ from those that were planned, transparent explanations for differences that affect the reliability of the study's results. If a reported analysis was performed based on an interesting but unanticipated pattern in the data, please be clear that the analysis was data-driven.

PLOS has an 'Inclusivity in Global Research' policy which aims to promote collaboration and inclusivity in global health research. You are required to complete PLOS’ questionnaire on inclusivity in global research and submit it with your revised paper. The policy and questionnaire can be found at https://journals.plos.org/plosone/s/best-practices-in-research-reporting.

Comments from Reviewers:

Reviewer #1: I thank the authors for their very detailed and helpful responses to my comments and to those of the other reviewers on this important paper.

Reviewer #2: All edits and suggestions addressed.

Reviewer #3: Thank you for addressing my comments but the responses raise a couple more questions:

1. Please ensure the rationale for selecting tests (e.g. parametric vs. non-parametric tests) are communicated through the text.

2. Thank you for clarifying the use of Spearman's correlation. It does assumes / examine only monotonic relationship. What i was wondering was if the relationship is more complicated than that. E.g. in Figure 5A, left panel, at a-axis around 2, the cluster of data points around y=0.5. I don't think there'd be any statistical methods that could test this because of the sample size is so small at that region - similarly these may as well be just random noise. However, I do think it is important to make sure the assumptions (i.e. monotonic relationship) is reported.

3. From the response to my question 4 in the first review, the authors implied that the selection of covariates are based on statistical significance which is not what is being recommended currently. Statistical inference for aetiological study should adjust based on a priori knowledge (common causes of exposure and outcome) rather than based on empirical p-values from univariate analysis. If the authors do not wish to reanalyse, please do acknowledge / discuss this.

[LINK]

---

## [Editor Report · Decision Letter 3]

20 Apr 2026

Dear Dr Beeson,

On behalf of my colleagues and the Academic Editor, Lorenz von Seidlein, I am pleased to inform you that we have agreed to publish your manuscript "Antibody fine specificity correlates with protection from malaria for the RTS,S vaccine in young African children: analysis of a phase IIb randomised control trial." (PMEDICINE-D-25-04372R3) in PLOS Medicine.

Please also note that we are requesting the following minor revisions:

Pleas add the URL for the NHMRC to the manuscript file and metadata.

Please correct this COI statement:". MP is a current employer and DW is a former employer of

ARTES Biotechnology GmbH ", revising 'employee' to 'employer', if correct. Corrections must be made to the manuscript and the metadata.

Please also provide an additional COI statement pertaining to the other authors (e.g. "The other authors declare that no competing interests exist.") in the manuscript and the metadata.

Please revise the title to :"Antibody fine specificity correlates with protection from malaria for the RTS,S vaccine in young African children: A post hoc analysis of a phase IIb randomised control trial"

In the Abstract, line 39: please correct phrasing to read:"and antibody responses in a large RTS,S..."

Line 46: insert 'we' to state "we found"

Line 52: there is one too many closed brackets here: "; p<0.001)))". Please correct.

Line 152-153: Please correct as follows: " including those contained in the RTS,S and R21 vaccines. We investigated IgG specificities in young African children vaccinated with RTS,S as part of a post hoc analysis of a phase IIb clinical trial."

Line 410-411: insert 'as' to read "referred to as Junction-repeat"

PRESS

Sincerely,

Alison Farrell, Ph.D.

Senior Editor

PLOS Medicine